# Self-similar growth of a bimodal laboratory fan

Pauline Delorme[1], Vaughan Voller[2], Chris Paola[2], Olivier Devauchelle[1], Éric Lajeunesse[1], Laurie Barrier[1], and François Métivier[1]

[1]Institut de Physique du Globe de Paris, Paris - Sorbonne Paris Cité, Université Paris Diderot, Paris, France
[2]Saint Anthony Falls Laboratory, University of Minnesota, Minneapolis, Minnesota, USA

*Correspondence to:* P. Delorme (pdelorme@ipgp.fr)

**Abstract.** Using laboratory experiments, we investigate the growth of an alluvial fan fed with two distinct granular materials. Throughout the growth of the fan, its surface maintains a radial segregation, with the less mobile sediment concentrated near the apex. Scanning the fan surface with a laser, we find that the transition between the proximal and distal deposits coincides with a distinct slope break. A radial cross section reveals that the stratigraphy records the signal of this segregation. To interpret
these observations, we conceptualize the fan as a radially symmetric structure that maintains its geometry as it grows. When combined with slope measurements, this model proves consistent with the sediment mass balance and successfully predicts the slope of the proximal-distal transition as preserved in the fan stratigraphy. While the threshold channel theory provides an order-of-magnitude estimate of the fan slopes, driven by the relatively high sediment discharge in our experimental system, the actual observed slopes are 3-5 times higher than those predicted by this theory.

## 1   Introduction

When a river leaves a mountain range to enter lowlands, it hits shallow slopes and loses valley confinement. This abrupt change causes it to deposit its sedimentary load into an alluvial fan (Bull, 1977; Rachocki and Church, 1990; Blair and McPherson, 1994; Harvey et al., 2005; Blair and McPherson, 2009). As the river builds this sedimentary structure, its bed rises above the surrounding land, and its channel becomes unstable. At this point, either the river erodes its banks to migrate laterally,
or, during a large flood event, it overflows, and in a process referred to as "avulsion", establishes a new course for its channel (Field, 2001; Slingerland and Smith, 2004; Sinha, 2009). In both cases, the river constantly explores new paths to fill up hollows in the deposit surface and preserve its radial symmetry. The resulting deposit acquires the conical shape which characterizes alluvial fans.

As the first sedimentary archive along the river's course, an alluvial fan records the history of its catchment (Hinderer, 2012).
Indeed, the geometrical reconstruction of a fan provides an estimate of its volume which, through mass balance, yields the average denudation rate of the catchment (Kiefer et al., 1997; Jayko, 2005; Jolivet et al., 2014; Guerit et al., 2016). Furthermore, when the river transports multiple grain sizes, it usually deposits the coarser sediment (gravel) near the fan apex, and the finer sediment (sand) at its toe. This segregation produces a gravel-sand transition front which moves forward and backward as the fan adjusts to external forcing. In radial cross section, this series of progradations and retrogradations appears as a boundary

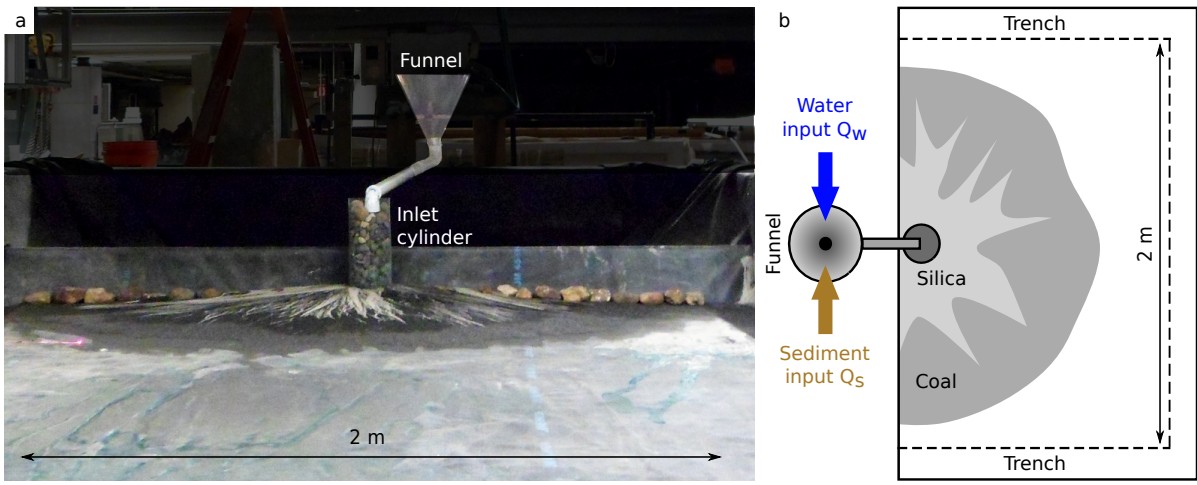

**Figure 1.** Experimental set-up. (a) Front-view picture. (b) Top-view representation.

between lithostratigraphic units, a pattern often interpreted as the signature of tectonic or climatic events (Paola et al., 1992a; Clevis et al., 2003; Charreau et al., 2009; Whittaker et al., 2011; Dubille and Lavé, 2015).

To interpret the morphology and stratigraphy of an alluvial fan, we need to understand how it translates the input signal (e.g., water and sediment discharges) into its own geometry (e.g., its size, downstream slope and stratigraphy). For instance, Drew (1873) observed that the lower the water discharge $Q_w$, the steeper the fan slope. More recent observations point at the influence of the sediment discharges $Q_s$ on the slope, often in the form of the ratio $Q_s/Q_w$. In general, the slope steepens when this ratio increases (Parker et al., 1998 a, b). At first sight, the shape of an alluvial fan is well approximated by a cone, but a closer look often reveals a steeper slope near the apex (Le Hooke and Rohrer, 1979; Blair, 1987; Blair and McPherson, 2009; Miller et al., 2014). Possible explanations for this include the decrease in sediment discharge caused by deposition (transport hypothesis), or the downstream fining of the sediment (threshold hypothesis) (Blissenbach, 1952; Rice, 1999; Stock et al., 2008; Miller et al., 2014). In practice, the variations of grain-size, slope and sediment discharge along a fan are correlated. When the sediment is broadly distributed in size, these variations are smooth, whereas a bimodal distribution generates a segmented fan (Bull, 1964; Williams et al., 2006).

Only seldom do field measurements allow us to separate the various parameters affecting the morphology of a fan, making it difficult to isolate their respective influence. One way around this problem is to use laboratory experiments, where small alluvial fans can be easily produced under well-controlled conditions (Schumm et al., 1987; Parker, 1999; Paola et al., 2009; Clarke, 2015). When water and sediment are injected onto the bottom of a tank, a deposit spontaneously forms around its inlet. The formation of this deposit is remarkably similar to that of natural fans; in particular a network of migrating and avulsing channels distributes radially the sediment across the fan surface. As the forcing parameters vary, the deposit responds by adjusting its morphology. Muto and Steel (2004), for example, showed that a base level fall induces upstream channel entrenchment, terrace abandonment, and fan progradation.

The sediment discharge $Q_s$ determines the growth rate of an experimental fan. Indeed, mass balance requires that the fan volume increase in proportion to the sediment input. Thus, as a consequence of the symmetry, the radius of the fan increases as $(Q_s t)^{1/3}$, where $t$ is the time elapsed since the beginning of the experiment (Powell et al., 2012; Reitz and Jerolmack, 2012). Avulsions occur more frequently as the sediment discharge increases, showing that the internal dynamics of an experimental fan adjusts to the forcings (Bryant et al., 1995; Ashworth et al., 2004; Clarke et al., 2010; Reitz and Jerolmack, 2012). This adjustment allows the fan to maintain its conical shape which, at first order and for a single grain size, is characterized by its slope only.

Even in simplified experiments (constant inputs, single grain size), there is no clear consensus about the mechanism by which a fan selects its own slope. Most investigators observed that a low water discharge, a high sediment discharge, and coarse grains all contribute to a steeper fan (Le Hooke and Rohrer, 1979; Clarke et al., 2010). However, the respective influence of water and sediment discharges on the slope remains debated. Whipple et al. (1998), Van Dijk et al. (2009) and Powell et al. (2012) hypothesized that the slope is a function of the dimensionless ratio $Q_s/Q_w$. In contrast, Guerit et al. (2014) propose that all three parameters act independently. In their experiment, a fan composed of uniform sediment grows between two parallel plates that confine it to the vertical plane. They found that the flow maintains the deposit surface near the threshold of motion. As a result, a lower water discharge causes the fan to steepen. The sediment discharge perturbs the fan profile only moderately, by steepening the slope in proportion to its intensity. As the sediment is deposited along the fan, the slope returns to its threshold value as it approaches the toe; the associated curvature in this process being proportional to the sediment input.

Accordingly, the downstream curvature of an alluvial fan composed of uniform sediment can be interpreted as a signature of spatial variation in sediment transport. However, one can wonder what happens when the fan is composed of non-uniform sediment? When the grain size is broadly distributed, downstream fining can also affect the fan profile. This phenomenon occurs in flume experiments, where large grains concentrate near the inlet (Paola et al., 1992b; Smith and Ferguson, 1996). In the experiment of Reitz and Jerolmack (2012), the fan builds its upper part out of large grains, and deposits the smaller ones near its toe. Consequently, the proximal slope is significantly steeper than the distal one, a signal whose form is similar to the curvature induced by deposition. We should also expect that this segregation would also appear in the fan's stratigraphy, a process that, to our knowledge, has not been previously investigated in laboratory experiments.

Here, we investigate the impact of a bimodal sediment on the morphology and stratigraphy of an alluvial fan. To do so, we generate a laboratory fan fed with a mixture of two granular materials (Sect. 2). Our experiment generates a segregated deposit, similar to the laboratory fan of Reitz and Jerolmack (2012). We first analyze its morphology, describing the growth of each part of the deposit independently . We then relate the spatial distribution of the sediment to the proximal and distal slopes (Sect. 3). Based on these observations, and appealing to the threshold-channel theory, we propose a geometrical model to describe the fan deposit (Sect. 4).

**Table 1.** Physical characteristics of the sediment. The measurement method is presented in Appendix A. The friction coefficient $\mu$ is the tangent of the angle of repose.

|  | Density | Grain size | |
|---|---|---|---|
|  | $\rho_s$ (kg m$^{-3}$) | $d_{50}$ ($\mu$m) | $d_{90}$ ($\mu$m) |
| Silica | $2650 \pm 50$ | 130 | 200 |
| Coal | $1500 \pm 50$ | 400 | 800 |

|  | Critical Shields | Friction coefficient |
|---|---|---|
|  | $\theta_c$ | $\mu$ |
| Silica | $0.25 \pm 0.02$ | $0.42 \pm 0.04$ |
| Coal | $0.19 \pm 0.008$ | $0.58 \pm 0.04$ |

**Table 2.** Experimental parameters for the five runs.

| Run | Water discharge | Sediment discharge | Silica fraction |
|---|---|---|---|
|  | $Q_w$ (L min$^{-1}$) | $Q_s$ (L min$^{-1}$) | $\phi$ |
| 1 | $2.6 \pm 0.1$ | $0.019 \pm 0.001$ | $0.5 \pm 0.05$ |
| 2 | $2.6 \pm 0.1$ | $0.045 \pm 0.001$ | $0.5 \pm 0.05$ |
| 3 | $2.6 \pm 0.5$ | $0.027 \pm 0.001$ | $0.25 \pm 0.02$ |
| 4 | $2.4 \pm 0.1$ | $0.027 \pm 0.001$ | $0.25 \pm 0.02$ |
| 5 | $2.6 \pm 0.1$ | $0.020 \pm 0.001$ | $0.8 \pm 0.08$ |

## 2 Experimental set-up

Producing experimental alluvial fans has become common in geomorphology (Schumm et al., 1987; Bryant et al., 1995; Whipple et al., 1998; Ashworth et al., 2004; Van Dijk et al., 2009; Clarke et al., 2010; Powell et al., 2012; Reitz and Jerolmack, 2012; Clarke, 2015). Here we use a setup similar to that of, for example, Whipple et al. (1998) to generate a radially symmetric fan over a horizontal basal surface (Fig. 1). In our experiments, however, a bimodal sediment mixture allows the fan to form a segregated deposit, visualized by color.

The tank we use to produce alluvial fans is 2 m-wide, and more than 5 m-long. Its bottom is covered with a black rubber tarpaulin. At the back of the tank, a 30 cm-high, vertical wall simulates the mountain front against which the fan leans. To prevent flow concentrations, large pebbles ($\sim$ 5 cm) are placed along the base of this back-wall. The three other sides are bounded by trenches to evacuate water (Fig. 1). It is noted that, even with these trenches, the surface tension maintains a 0.5 cm-deep sheet of water over the base of the tank. Assuming that this standing water affects only the base of the fan, we find

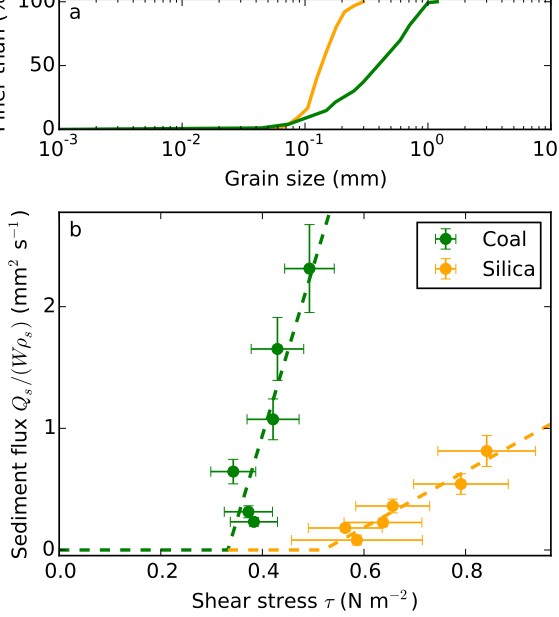

**Figure 2.** (a) Cumulative density function of the grain size. Orange: silica, green: coal. (b) Transport laws. Volumetric flux per unit width, as a function of dimensional shear stress. Dashed lines correspond to Eq. (A3) fitted to the data (method in Appendix A, coefficients in Table 1).

that it represents less than 1% of its volume. Based on this simple calculation, we hereafter neglect its influence in our analysis and interpretation.

To ensure constant inputs of water and sediment into the experiment, we use a constant-head tank to supply the water, and an Archimedes screw to supply the grains. The fluxes of water and sediment merge in a funnel, which directs them toward the tank. Before reaching the fan, water and sediment flow through a 10 cm-wide, wire-mesh cylinder filled with pebbles. This device reduces the water velocity and homogenizes the mixture (Fig. 1).

The mixture we used is composed of black coal and white silica grains, the colors of which are easily distinguished. The coal grains are larger and lighter than the silica grains (Table 1). To quantify the mobility of these grains, we measure their respective transport laws in independent experiments (Appendix A). We find that both transport laws, for pure coal and pure silica, exhibit an unambiguous threshold, below which there is no transport (Fig. 2). This threshold is about $0.34\,\mathrm{N\,m^{-2}}$ for coal, and $0.52\,\mathrm{N\,m^{-2}}$ for silica. Beyond this threshold, the sediment flux appears proportional to the distance to threshold, with a proportionality constant of $2.4\,10^{-5}\,\mathrm{m^2\,s^{-1}}$ for coal and $4.8\,10^{-6}\,\mathrm{m^2\,s^{-1}}$ for silica. As a result, the same shear stress $\tau$ induces a larger flux of coal than silica, at least when the two species are unmixed. In other words, despite their larger size, the coal grains are more mobile than the silica ones. This, of course, is due the first being lighter than the latter. To formalize this density-induced reversal of mobility, we need to introduce the Shields parameter $\theta$, which is the ratio of the shear stress over

the grain's weight (Shields, 1936):

$$\theta = \frac{\tau}{(\rho_s - \rho)gd_s}, \tag{1}$$

where $\rho$ is the density of water, $\rho_s$ is the density of sediment, $g$ is the acceleration of gravity, and we approximate the grain size $d_s$ with its median value $d_{50}$. For our sediments, the denominator in Eq. (1) is larger for silica than for coal, indicating that the density difference prevails over grain size to govern the mobility of our grains. This is in contrast to the experiments of Reitz and Jerolmack (2012), where the mobility difference is driven by grain size. When expressing the threshold for transport in terms of the Shields parameter, we find $\theta_c = 0.19$ for coal, and $\theta_c = 0.25$ for silica (Table 1). These values reinforce the mobility contrast induced by density.

When different grains are mixed, the shear stress exerted on each species depends on the mixture composition (Wilcock and Crowe, 2003; Houssais and Lajeunesse, 2012). The shear stress required to move the larger grains in a mixture is lower than for large grains alone, because the smaller ones cause them to protrude into the fluid. Conversely, small grains in a mixture require a higher shear stress because they are shielded from the flow by neighboring large grains (Einstein, 1950). For grains of different densities but uniform size, exposure and hiding are negligible (Viparelli et al., 2015). There exist no universal transport law accounting for all these phenomena, and deriving an empirical one for our mixture would be a daunting task. We thus use the transport laws of Fig. 2 to account for differential transport and estimate the mobility of our grains, although this is certainly a rough approximation. If it holds, at least qualitatively, we expect the rivers that build our experimental fans to segregate the sediment based on grain mobility, by depositing silica while transporting coal further downstream.

An experimental run begins with an empty tank. When the mixture of water and sediment reaches the horizontal bottom of the tank, it forms a half-cone deposit. Initially, a sheet flow spreads uniformly over this sediment body. After a few minutes, the flow confines itself into distinct, radial, channels (typically five or six). All these channels appear to transport sediment simultaneously. The experiment of Reitz and Jerolmack (2012) also produced about 5 channels, although only one of them was active at a time. Both configurations occur in the field (Weissmann et al., 2002; Hartley et al., 2010). In our experiments, bedload appears as the dominant transport mode, although a small amount of fine coal is suspended, and gets deposited on the banks. The width of our channels varies between about 1 and 2 cm. Assuming they share the total water discharge evenly, the typical Reynolds number of their flow is above 500, suggesting that, most of the time, they are turbulent. They avulse regularly to maintain the radial symmetry of the fan. During an avulsion, overbank flow occurs temporarily, a phenomenon also observed by Bryant et al. (1995) and Reitz and Jerolmack (2012). Our experiment stops when the deposit reaches the sides of the tank, typically after 3 to 4 hours.

As it grows, the fan deposits the silica grains upstream of the coal grains. Accordingly, the apex of the fan is mostly composed of silica, whereas coal constitutes most of its toe. The boundary between the two types of sediment follows the path of channels, thus adopting a convoluted shape. To explore the influence of the sediment composition on the morphology of the fan, we varied $\phi$, the volumetric proportion of silica in sediment mixture, from 25% to 80% over five experiments (Table 2).

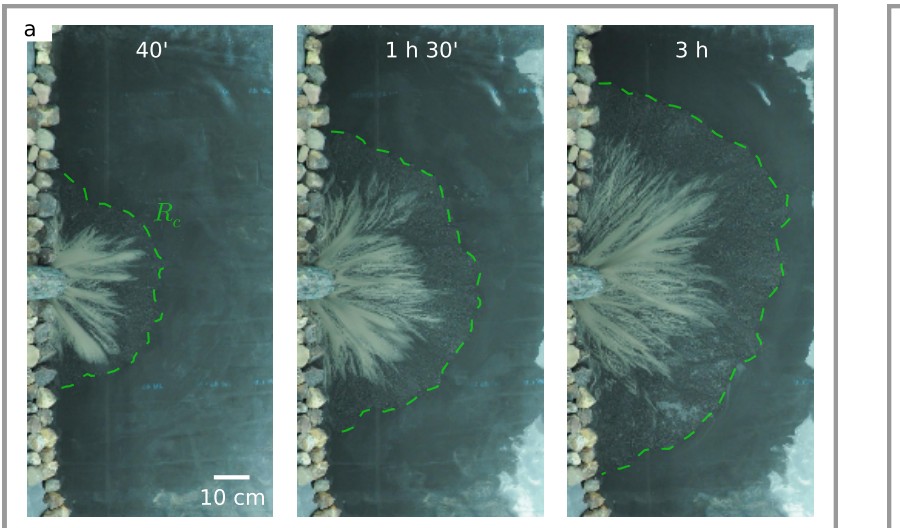
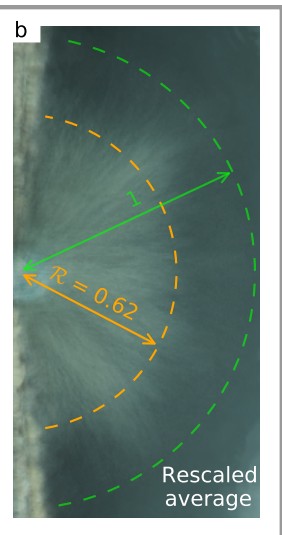

**Figure 3.** Top-view pictures of an experimental fan (run 2). (a) Time evolution. Green dashed line indicates fan toe, $R_c$. (b) Average of rescaled pictures. The 26 pictures are each 10-minutes apart. Dashed lines indicate silica-coal transition (orange) and fan toe (green). After rescaling, the fan length is one. Transition between silica and coal occurs at dimensionless distance $\mathcal{R}$ from apex.

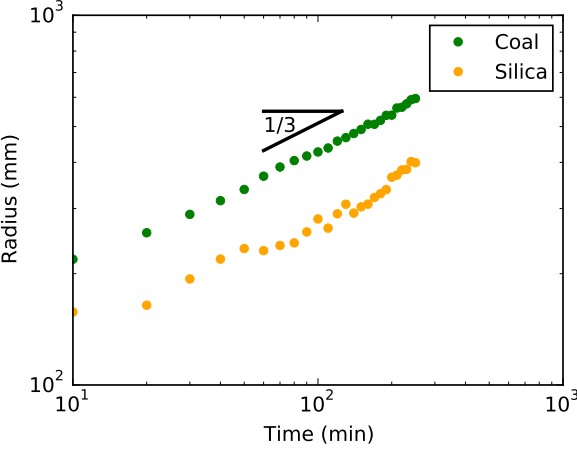

**Figure 4.** Evolution of the radial fronts of the silica (orange) and coal (green) deposits in run 2.

## 3 Self-similar growth of a segmented fan

During each run, we track the evolution of the fan surface with a camera (Nikon D90 with a wide-angle lens Nikon AF DX Fisheye-Nikkor 10.5 mm f/2.8G ED) fixed above the center of the tank. We record an image every minute (Fig. 3a). The exact location of the boundary between silica and coal varies significantly during a run. For a run, however, this boundary appears at

a constant location relative to the fan length. This fraction depends on the composition of the sediment mixture (Table 3). To confirm this observation, we manually locate the fan toe on 26 pictures, 10 minutes apart from each other (Fig. 3a). From these individual measurements, we estimate the average radius $R_c$ of the fan with an accuracy of about 6% on each picture. We then

rescale each picture with the corresponding value of $R_c$, thus normalizing the size of the fan to one. Finally, we average all the normalized pictures of an experimental run (Fig. 3b). By construction, the average picture shows a fan of radius one. It also confirms that the fan is radially symmetric, and reveals a somewhat blurred but localized transition between the silica and coal deposits. This observation suggests that the fan preserves the spatial distribution of coal and silica as it grows.

To verify the self-similarity of the fan growth, we analyze the evolution of its geometrical properties. To do so, we manually

locate the silica-coal transition and the fan toe (Fig. 3a). We then calculate the average distance $R_s$ from the apex to the transition. The boundary of the silica deposit being more convoluted than the toe, the standard deviation of $R_s$ is about 19%. Both distances increase in proportion to the cube root of time (Fig. 4). Following Powell et al. (2012) and Reitz and Jerolmack (2012), we interpret this observation as a direct consequence of mass balance. Indeed, the total mass $M$ of the deposit increases linearly with time:

$$M = Q_{s,m}\, t \tag{2}$$

where $Q_{s,m}$ is the total mass flux of sediment. To express this relation in terms of volumes, we need to measure the packing fraction $\lambda$ of our sediment mixture. In general, this quantity depends on the composition of the mixture. To estimate it, we measure the packing fraction of pure silica, of pure coal, and of a 50% silica-coal mixture (red dots, Fig. 5). The three values are similar, with a mean of $55\% \pm 1\%$. Accordingly, we approximate the packing fraction of the entire deposit with this value,

regardless of the composition of the sediment mixture. This approximation introduces an error of less than 5%. We now define the volume discharge of sediment $Q_s$, such that

$$Q_{s,m} = Q_s(1 - \lambda)\,(\phi \rho_s + (1 - \phi)\rho_c) \tag{3}$$

and substitute $Q_s$ for $Q_{s,m}$ in Eq. (2). The mass balance then reads

$$V = Q_s\, t \tag{4}$$

where $V$ is the total volume of the deposit. In a self-similar fan, any distance scales like the cube root of the fan volume; in particular, both $R_c$ and $R_s$ increase in proportion to $(Q_s t)^{1/3}$. Our experimental fans conform to this scaling, thus supporting the hypothesis of a self-similar growth. A direct consequence of this self-similarity is that the relative location of the transition, defined by the ratio $\mathcal{R} = R_s\,/\,R_c$, remains constant throughout growth ($\mathcal{R} = 0.62 \pm 0.04$ for run 2, other runs are presented in Table 3, Figs.3b and 4). This self-similarity means that, as it grows, the fan preserves its structure, which can therefore be

extrapolated from the final deposit.

A few minutes after the experiment stops, all the surface water has drained away from the fan, leaving the entire deposit emergent. At this point, we scan the deposit's surface with a laser to measure its topography (OptoEngine MRL-FN-671, 1 W, 671 nm). A line generator converts the beam into a laser sheet (60° opening angle, 1 mm thick), the intersection of which with

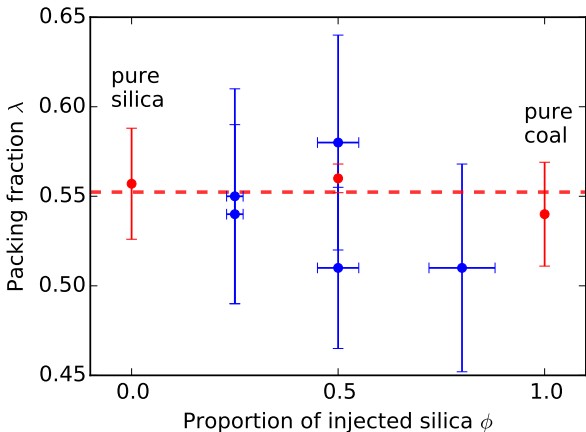

**Figure 5.** Packing fraction of the deposit as a function of the composition of the sediment mixture. Blue dots calculated from experiment. Red dots measured independently. The red dashed line is the mean packing fraction measured independently.

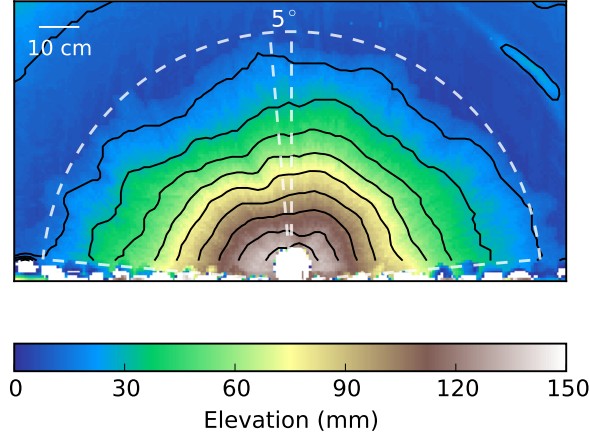

**Figure 6.** Digital elevation model of an experimental fan (run 2). Black lines: elevation contours 15 mm apart from each other. White dashed lines indicate the bounds used for averaging (only two sample radii 5 degree apart are represented for clarity).

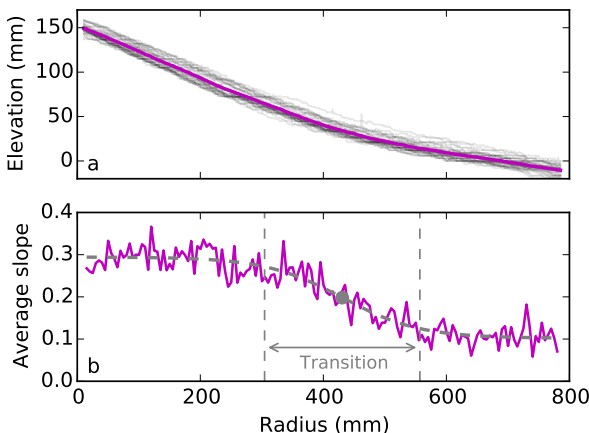

**Figure 7.** (a) Fan profiles at different angles (run 2). Gray: individual profiles; magenta: average profile. (b) Average downstream slope (magenta). Fitted hyperbolic tangent (dashed gray). Inflection point (gray dot) and boundary of the transition area (vertical dashed gray line).

the fan surface is recorded by a camera attached to the laser, about 2 m above the tank bottom (Sick Ranger E50, 12.5 mm lens). The precision of the measurement is better than 1 mm in every direction.

Using the digital elevation model (DEM) of our experimental fan, we compute the final volume of our fans to check the total packing fraction of the deposit (blue dots, Fig. 5). Despite some dispersion, we find that the packing fraction of our deposit is about $54\% \pm 2\%$, close to the value estimated independently.

The elevation contours of the DEM are well approximated by concentric circles, another indication of radial symmetry (Fig. 6). This property suggests that we can compute the radially-averaged profile of the fan with minimal loss of information (Reitz and Jerolmack, 2012). To do so, we interpolate the DEM along 34 radii, 5° apart from each other, at the end of each run (Fig. 6). For each run, the resulting profiles are similar to each other, and differ from the mean by less than 7% (Fig. 7a). The average fan profile is steeper near the apex than at the toe and can be approximated by two segments of uniform slope. Natural fans sometimes feature a similarly segmented profile (Bull, 1964; Blair and McPherson, 2009; Miller et al., 2014). When we plot the downstream slope of this average profile as a function of the distance to the apex, the transition appears as a decreasing sigmoid curve (Fig. 7b). To evaluate the location of the transition, and the extension of the transition zone, we fit a hyperbolic tangent to the slope profile (Fig. 7b for run 2, other runs in Table 3). For run 2, we find that the slope plateaus to a value of about 0.29 near the apex, and to about 0.10 near the toe. We define the location of the transition as the inflection point of the sigmoid, which occurs at $55\% \pm 9\%$ of the total fan length (Fig. 7b). The slope thus breaks where the sediment turns to coal, suggesting that these transitions are closely related (Fig. 3, $\mathcal{R} \approx 0.62 \pm 0.04$). The location of the transition depends on the mixture composition (Table 3). We now define the extension of the transition zone as the characteristic length of the sigmoid. For run 2, we find that the transition between the two segments of the fan occurs over a length of 32% of the total fan length. This value is almost independent of the sediment mixture (about $30\% \pm 3\%$ on average for all runs). Miller et al. (2014) found a comparable value (about 22%) for natural and laboratory fans.

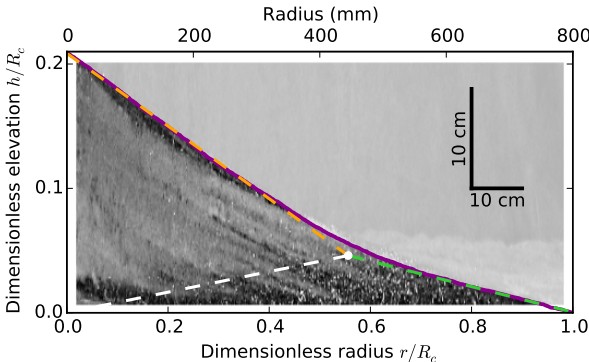

**Figure 8.** Average radial profile (magenta line), superimposed on radial cross section, for run 2. Dashed lines: slope of the silica (orange) and coal (green) deposits. White dashed line indicates silica-coal transition. Scale is for the picture. $h$ and $r$ are respectively the fan elevation and the distance to the apex.

To investigate the relation between the slope break and the silica-coal transition, we now turn our attention to the internal structure of the deposit. After the water and sediment supplies have been switched off, the fan remains intact, and we can cut it radially to reveal a vertical cross section (Fig. 8). Silica and coal appear segregated, in accordance with the top-view pictures of the fan (Fig. 3), and with the experiments of Reitz and Jerolmack (2012). Silica concentrates near the apex, in the upper part of the deposit, whereas coal concentrates at the fan toe. The location of the silica-coal transition fluctuates, and generate an intricate stratigraphy that combines segregation at the fan scale, and stratification near the transition. The transition zone shows alternating layers of silica and coal, which extend over about one third of the cross-section area. In natural fans, such stratifications result from fluctuations of the sediment and water discharges, but this mechanism cannot be invoked in our experiments (Paola et al., 1992a; Clevis et al., 2003; Whittaker et al., 2011). Dry granular flows can also generate a similar pattern (Makse et al., 1997b, a). In our case, the succession of channel avulsions is another possible mechanism. Our observations do not allow us test these hypotheses.

The surface of the cross section resembles the average profile of Fig. 7a. Indeed, when superimposed, the two lines become virtually indistinguishable, with the slope break occurring near the transition between silica and coal (Fig. 8). Neglecting the span of the transition, we may approximate the average profile by fitting two straight lines to it. The proximal line joins the apex to the transition (slope = 0.29), and the distal line joins the transition to the toe (slope = 0.10). The two lines intersect at 56% of the deposit length. Finally we define the transition line, which joins this intersection to the origin and passes through the alternating stratigraphic layers in the transition zone. The transition line thus divides the deposit into two imbricated wedges, with the more mobile sediment (coal) lying below the less mobile one (silica). The upward migration of the sand-coal transition in the deposit section reflects the outward growth of the transition accompanied by net deposition. In the next section, we formalize this interpretation in the context of self-similar growth, and combine it with mass balance to understand how the fan builds its deposit.

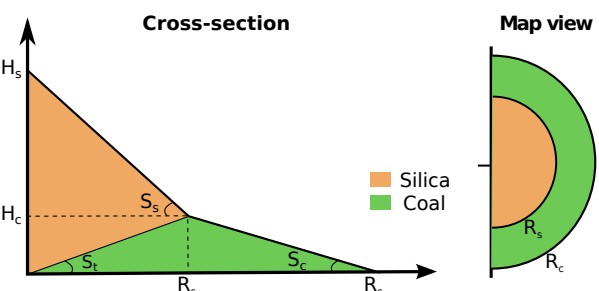

**Figure 9.** Representation of an alluvial fan (template). Silica: orange; coal : green.

## 4  Mass balance

Based on our laboratory observations, we propose a first-order geometrical model of an alluvial fan fed with a bimodal mixture of sediments. We consider a radially symmetric structure, which grows by expanding itself without changing its geometry. A consequence of these assumptions is that the geometry of the fan, at any time, is entirely determined by a fixed, two-dimensional template of its cross section (Fig. 9). The simplest possible template consists of two triangles with a common side. The proximal triangle defines the geometry of the silica deposit, and the distal one represents the coal deposit. Three dimensionless parameters define this template: the proximal slope $S_s$, the distal slope $S_c$ and the relative location of the transition $\mathcal{R} = R_s/R_c$.

The geometry of the template sets the proportion of silica and coal in the deposit. As a consequence, mass balance relates the three parameters that define the fan template to the composition of the sediment mixture injected in the experiment, $\phi$. Indeed, since the sediment discharge is constant, and assuming the deposit is fully segregated and the packing fraction is constant, we should have

$$\frac{V_s}{V_s + V_c} = \phi, \tag{5}$$

where $V_s$ is the volume of silica in the deposit, and $V_c$ that of coal. For a self-similar fan, this relationship holds at any time.

The silica deposit is composed of two half-cones sharing their base. Its volume reads

$$V_s = \frac{\pi}{6} R_s^2 H_s, \tag{6}$$

where $H_s$ is the elevation of the fan apex. To calculate the volume of coal in the deposit, we first evaluate that of a truncated half cone with slope $S_c$, radius $R_c$, and height $H_c$ (the elevation of the transition). We then withdraw the volume of the lower cone of the silica deposit. The resulting volume reads

$$V_c = \frac{\pi}{6} \left( R_c^2 + R_c R_s \right) H_c. \tag{7}$$

The proximal and distal slopes are simply those of the corresponding right triangles:

$$S_s = \frac{H_s - H_c}{R_s} \quad \text{and} \quad S_c = \frac{H_c}{R_c - R_s}. \tag{8}$$

**Table 3.** Geometrical characteristics of the experimental fans, measured at the end of each run. The errors on $\mathcal{R}$ are due to fluctuations of the silica-coal transition.

| Run | Slope ratio $\mathcal{S}$ | Transition location $\mathcal{R}$ | $S_t/S_s$ $\mathcal{S}_t$ |
|-----|-----------|---------------------|-----------|
| 1 | $3 \pm 0.3$ | $0.56 \pm 0.07$ | $0.37 \pm 0.08$ |
| 2 | $2.9 \pm 0.1$ | $0.55 \pm 0.09$ | $0.36 \pm 0.08$ |
| 3 | $4 \pm 0.6$ | $0.41 \pm 0.1$ | $0.57 \pm 0.12$ |
| 4 | $4.6 \pm 0.6$ | $0.39 \pm 0.1$ | $0.61 \pm 0.13$ |
| 5 | $3.3 \pm 0.3$ | $0.83 \pm 0.2$ | $0.12 \pm 0.03$ |

Using the four above equations, we finally relate the composition of the sediment mixture to the geometry of the fan, as a function of the slope ratio and the transition location:

$$\phi = \frac{(1-\mathcal{S})\mathcal{R}^4 - \mathcal{S}\mathcal{R}^3 - \mathcal{R}^2}{(\mathcal{R}+1)\left((1-\mathcal{S})\mathcal{R}^3 - 1\right)}, \tag{9}$$

where we have defined the ratio of proximal slope to distal slope $\mathcal{S} = S_s/S_c$. Equivalently we may express the composition of the sediment mixture as a function of the slope ratio and the slope of the transition:

$$\phi = \frac{1-\mathcal{S}_t}{1 + \mathcal{S}_t\left((\mathcal{S}\mathcal{S}_t)^2 + 3\mathcal{S}\mathcal{S}_t + 2\right)}, \tag{10}$$

where we have defined the ratio of transition slope to proximal slope $\mathcal{S}_t = S_t/S_s$.

If the template is a reasonable representation of the fan geometry, the location and the slope of the transition and the two surface slopes of the deposit should adjust to the composition of the sediment input, according to Eqs. (9, 10). To evaluate this model, we measure the geometry of the fan at the end of every experimental run (Table 3). Using the radially averaged profile we first fit, using a linear regression, the proximal and distal slopes and calculate their ratio. Then, we estimate the location of the transition using the position of the inflection point (Sect. 3). We find that, for all runs, the proportion of silica in the deposit, as deduced from our measurements through Eqs. (9, 10), matches the composition of the sediment mixture (Fig. 10).

At first order, we can thus represent our experimental fan as a radially symmetric, fully segregated structure which preserves its shape as it grows. These features determine the dynamics of the fan, and the geometry of its deposit. This model, however, involves two free parameters: the proximal and distal slopes. These are selected by the fan itself, by a mechanism that remains to be understood. Each deposit is built by a collection of channels, which select their own slope according to the composition of the bed, and to their sediment and water discharges. On the DEM of our experimental fans, the channels are virtually invisible, showing that their downstream slope is that of the fan (Fig. 6). It is thus reasonable to assume that the deposit inherits the slope of the channels that build it. The way a river selects its morphology is still a matter of debate, but it has been recently pointed out that most laboratory rivers, including those flowing over an experimental fan, remain near the threshold for sediment transport (Reitz and Jerolmack, 2012; Seizilles et al., 2013; Reitz et al., 2014; Métivier et al., 2016). Assuming a channel is exactly

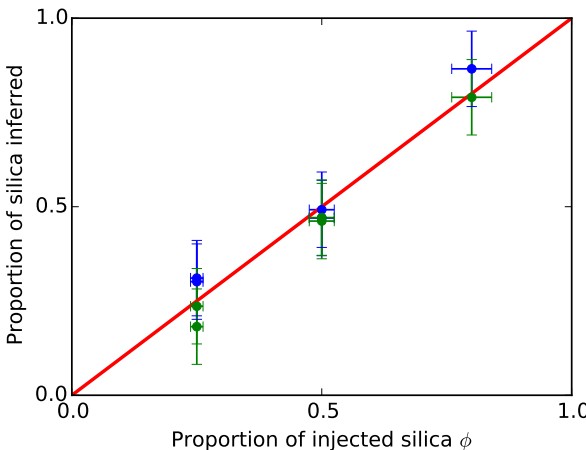

**Figure 10.** Proportion of silica inferred from the geometry of the deposit, after Eq. (9) (blue) and after Eq. (10) (green), as a function of the composition of the sediment input. Red line: perfect agreement.

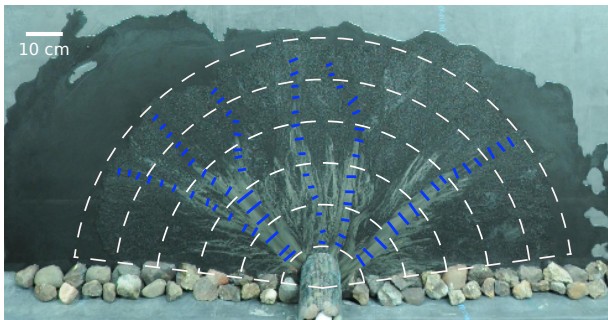

**Figure 11.** Top-view of an experimental fan superimposed with measurement bins (white), and channels cross sections (blue).

at threshold yields a theoretical relationship between its water discharge and its slope (Glover and Florey, 1951; Henderson, 1961). Could this theory inform us about the slope of our fans?

10    Returning to our experimental fans, we find them enmeshed in a collection of channels flowing radially (Fig. 11). These channels sometimes bifurcate downstream, but do not recombine as they would in a braided river. We would like to compare their slope to the prediction of the threshold-channel theory. Unfortunately, our experimental setup does not allow us to measure the water discharge of individual channels. If the flow distributes itself evenly among the channels, though, we can approximate their individual discharges to a fraction of the total discharge. To evaluate this approximation, we now analyze top-view pictures

15    of our developing fans (about 15 pictures per run). We first divide the surface of each fan into five concentric bins, where we count the active channels and measure their widths (at least two cross sections per channel and per bin, Fig. 11). We then average the number of channels, and their width, over experimental runs. The resulting quantities depend on the time of their

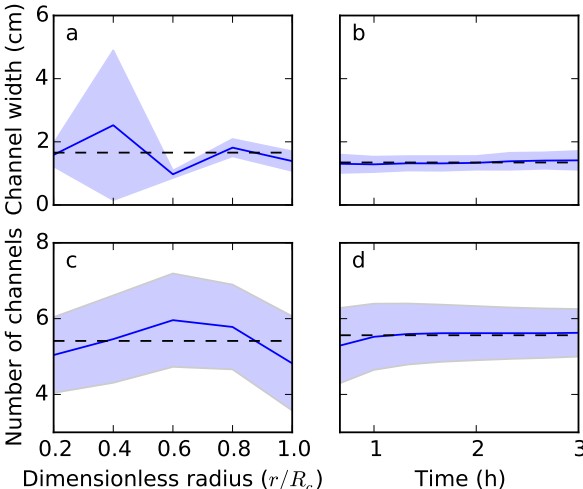

**Figure 12.** Evolution of active channels for all the runs. Channel width as a function of the dimensionless radius (a) and time (b). Number of channels as a function of dimensionless radius (c) and time (d). Black dashed line: average. Shaded area: variability over experimental runs.

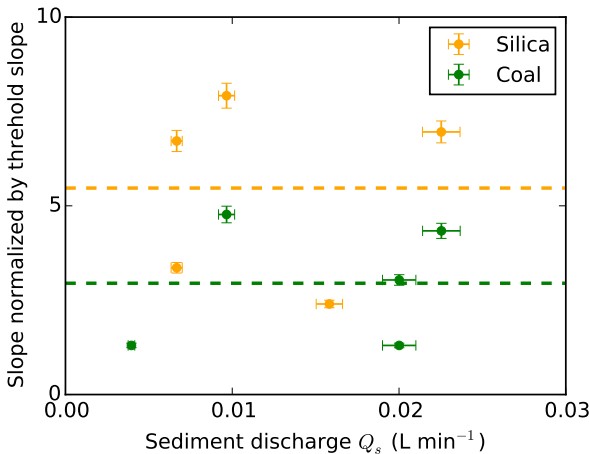

**Figure 13.** Slope normalized by the threshold slope, calculated with Eq. (11), as a function of the sediment discharge. Dashed lines: average slopes.

measurement, and on the distance from the apex, $r$. Further averaging over time yields radius-dependent quantities, whereas averaging over distance yields time-dependent quantities (Fig. 12).

When plotted as a function of radius, the width of the channels varies between about 1 and 2.5 cm, with no clear trend (Fig. 12a). The variability of the width is much larger in the proximal part of the fan than in its distal part. When plotted as a function of time, we find that the width is more consistent, with a relative variability of about 10% around a mean value

of 1.3 cm (Fig. 12b). Overall, the channels appear reasonably homogeneous in size, suggesting that they share the total water discharge evenly.

The number of channels $n_c$ varies between 5 and 6 across the fan (Fig. 12c). As expected for a radially oriented structure, we count fewer channels near the apex. We also find fewer channels near the toe, although the poor color contrast of the coal-dominated areas probably bias our count. This variability compares with the disparity we observe between runs. The number of channels is nearly constant over time (Fig. 12d). Hereafter, we choose $n_c = 5.5$, and divide the total water discharge accordingly.

We now wish to compare the slope of our experimental fans with the threshold theory, applied to the characteristic channel defined above. This theory assumes that the combination of gravity and flow-induced shear stress maintains the channel bed at the threshold of motion (Glover and Florey, 1951; Henderson, 1961; Seizilles et al., 2013). As a result, the width, depth and slope of the channel are set by its water discharge. In particular, according to the simplest version of this theory (Devauchelle et al., 2011; Gaurav et al., 2015; Métivier et al., 2016), the equilibrium slope reads

$$S_H = \left( g\,\mu^3 \left( \frac{\theta_c}{\mu}\, \frac{\rho_s - \rho}{\rho}\, d_s \right)^5 \right)^{1/4} \sqrt{\frac{2^{3/2}\,\mathcal{K}(1/2)n_c}{3\,C_f\,Q_w}}, \tag{11}$$

where $\mu$ is Coulomb's coefficient of friction (Table 1), $\nu = 10^{-6}$ m$^2$s$^{-1}$ is the kinematic viscosity of water, $\mathcal{K}(1/2) \approx 1.85$ is the elliptic integral of the first kind, and $C_f$ is Chézy's coefficient of fluid friction. The Chézy coefficient $C_f$ depends on the bed roughness and the flow Reynolds number. For simplicity, we approximate $C_f$ with a constant value of 0.02 (Moody, 1944; Chow, 1959). Since we imposed the same water discharge during all experimental runs, and found the number of channels $n_c$ to be relatively constant, the slope corresponding to the threshold theory depends on the sediment only. We find $S_{Hs} \approx 0.042$ for silica, and $S_{Hc} \approx 0.023$ for coal, using Eq. (11).

Intuitively, we expect that, all things being equal, the fan slope increases with sediment discharge. Previous observations support this intuition, but there is no consensus yet about its physical origin, which involves the response of a single channel to sediment transport and its destabilization into multiple threads (Whipple et al., 1998; Ashworth et al., 2004). We do not find any correlation between sediment discharge and slope in our experiment (Fig. 13). Even after normalizing our measurements according to the threshold theory, the data points appear segregated according to the sediment species: the mean slope of the silica deposit is about $S_s/S_{Hs} = 5.6 \pm 2.0$, whereas we find $S_c/S_{Hc} = 2.9 \pm 1.5$ for coal ($S_s \approx 0.23$ and $S_c \approx 0.068$). The surface slopes of the two fan segments are thus significantly higher than predicted by the threshold theory.

A possible cause for this departure from the threshold channel could be the bimodal mixture we use. To assess this hypothesis, we have produced an experimental fan with pure silica ( $Q_s \approx 0.014$ L min$^{-1}$, $Q_w \approx 2.6$ L min$^{-1}$ ). We found that, like its bimodal counterparts, its slope was approximately five times higher than predicted by the threshold-channel theory ($S_s = 0.2$). Another possible explanation is the infiltration of surface water into the deposit. Indeed, based on Eq. (11), a lower water discharge induces a steeper channel. Measuring this leakage would be experimentally challenging. Finally, the breakdown of the threshold-channel theory could result from sediment transport, since active channels must be above threshold (Whipple et al., 1998; Guerit et al., 2014). In their one-dimensional experiment, Guerit et al. (2014) have shown that the higher the

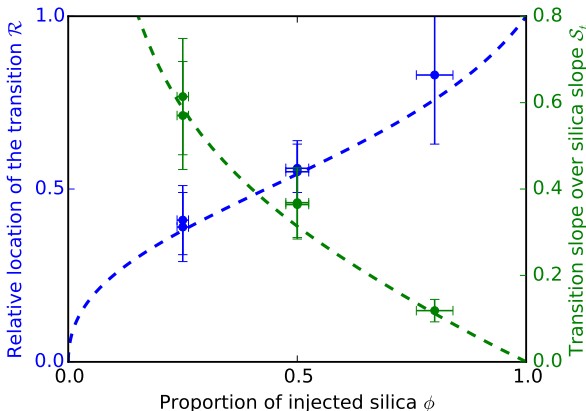

**Figure 14.** Relative position of the transition $\mathcal{R}$ (blue) and dimensionless transition slope $\mathcal{S}_t$ (green), as a function of the composition of the sediment input. Dot: experimental measurements. Dashed line: Eqs. (9, 10) with $\mathcal{S} = 3.4$.

sediment input in their experiment, the more slope departs from its threshold value. Again, we cannot evaluate quantitatively
this hypothesis in our experiments.

The proximal and distal slopes seem independent from sediment discharge (Fig. 13). For lack of a physical interpretation, we now treat this observation as an empirical fact, and attribute a fixed value to the ratio of proximal slope to distal slope: $\mathcal{S} = S_s/S_c = 3.4 \pm 1.0$. Substituting this value in Eqs. (9, 10), the mass balance relates, without any additional parameters, the composition of the sediment mixture to the location and the slope of the transition (Fig. 14). Despite significant uncertainties,
which probably reflect the rudimentary mass balance we used, our observations agree with this semi-empirical relationship.

In principle, one could use Fig. 13 to infer the composition of the sediment input from the geometry of the deposit. This method, however, relies on the value of the slope ratio $\mathcal{S}$, which we have fitted on our observations. A more comprehensive theory should explain how a bimodal fan spontaneously selects the value of this ratio.

## 5   Conclusion

Using a laboratory experiment, we generated alluvial fans fed with a bimodal sediment. Five or six active channels deposit their sediment load to form a radially-symmetric fan. The heavier sediment (silica) concentrates around the apex, whereas the lighter one (coal) get deposited near the toe. The location of silica-coal transition fluctuates over about 30% of the total fan length. A radial cross section of the deposit reveals a similar segregation: two superimposed triangles make up the stratigraphy of the fan. The lowest triangle is mostly coal, whereas the upper one, located near the apex, is mostly silica. The transition between
the two parts of the fan fluctuates to produce strata, which extend over 30% of the total fan length. As a first approximation, we may represent this transition with a straight line, and treat the fan structure as two imbricated deposits. Combining this geometric model with mass balance, we find that the fan preserves this structure as it grows, with a precision of about 15%. This observation suggests that our laboratory fans act essentially as sieves, which segregates the sediment they are fed with.

This process controls the geometry of the resulting deposit. As a consequence, we can use the final geometry of our laboratory fans to infer the composition of the sediment input. In practice, a top-view picture of the deposit suffices to do so. Alternatively, measuring the slope of the transition in the stratigraphy, even if the latter is incomplete, also suffices.

Natural fans often exhibits a sharp transition from gravel to sand (Blair and McPherson, 2009; Miller et al., 2014). Like in our experiments, this front divides the fan profile into two segments. The proximal segment, composed mainly of gravel, is steeper than the distal one, composed mainly of sand. Bull (1964) and Blair (1987) found natural fans featuring three segments bounded by two successive transitions. Again, the size of the deposited sediment changes abruptly at each front. These observations suggest that the segregation mechanism at work in our experiment can repeat itself to generate nested deposits. A natural extension of our work would be to enrich the sediment mixture with additional grain sizes (or densities) to produce fans with multiple segments. We would expect these fan to sort sediment species based on their mobility, and reduce their slope downstream, as observed on the surface of many natural fans (Stock et al., 2008). In other words, the structure of an alluvial fan should reflect the composition of its sediment input. For instance, in principle, one could infer the grain-size distribution of the sediment input from a DEM of the fan.

In practice, however, secondary processes such as weathering, runoff, and aeolian erosion reworks the surface of most natural fans, thus hampering our ability to infer their history from their present state (de Haas et al., 2014). To circumvent this issue, one can either reconstruct geometrically the paleosurface of the fan, or use its stratigraphy. Indeed, even partial access to the internal structure of the fan could reveal the slopes of the transitions in the stratigraphy, and thus the grain-size distribution of the input.

In our experiments, the inputs of water and sediment were constant. In general, this is not true for natural fans, and the interpretation we propose here does not apply in its present, oversimplified, form. The self-similar model we propose here thus cannot account for climatic and tectonic signals. However, the fundamental hypothesis upon which it relies, namely that the fan sorts the sediments based on their mobility and adjusts its own slope accordingly, might still hold when the inputs fluctuate. If so, our geometrical model might be extended to account for these fluctuations. This is the subject of present work.

Our experiments also suggest that the process by which an alluvial fan distributes grain sizes in its deposit, although a primary control on its structure, may not be the most puzzling component of its machinery. The way it selects its slope remains a challenging problem, which we have circumvented here by fitting a parameter to our observations (Le Hooke and Rohrer, 1979; Whipple et al., 1998; Stock et al., 2008; Van Dijk et al., 2009; Powell et al., 2012; Guerit et al., 2014). Indeed, the threshold theory can only provide us with a first-order estimate for the slope of a channel. We need to understand how a channel adjusts its slope to its sediment load. Recent investigations have shown that, provided the sediment discharge is low enough, one can produce stable active channels in laboratory experiments (Seizilles et al., 2013; Métivier et al., 2016). If this method works for a laboratory fan as well, it might generate a single-channel fan. This would be a simpler experimental tool to investigate the relationship between the slope of a fan and the intensity of its sediment input.

## 5 Appendix A: Transport law

To calibrate the transport laws of our sediments, we use an independent set-up similar to that of Seizilles et al. (2014). The flow is confined between two Plexiglas panels separated by a 3.2 cm-wide gap in which we inject water and sediment at constant rate. Once the experiment has reached equilibrium, typically ten to twenty hours after it started, we measure the slope of the water surface $S$ to estimate the shear stress $\tau$. Since the Reynolds number is below 500 in our flume, we may assume that the flow is laminar. The shear stress acting on the sediment thus follows Poiseuille's law:

$$\tau = \rho(Sg)^{2/3}\left(\frac{3Q_w\nu}{W}\right)^{1/3}, \tag{A1}$$

where $W$ is the width of the gap, and $\nu$ the viscosity of water. We then calculate the Shields parameter, which represents the ratio of the flow-induced shear stress $\tau$ to gravity:

$$5 \quad \theta = \frac{\tau}{(\rho_s - \rho)gd_s}, \tag{A2}$$

and calibrate the transport law (Fig. 2). We find that below a critical value $\theta_c$, which correspond to a critical shear stress $\tau_c$, the sediment flux vanishes. Above this threshold, the flux appears proportional to the departure from the critical Shields parameter:

$$\frac{Q_s}{W} = q_0(\theta - \theta_c), \tag{A3}$$

10 where $q_0 = 4.8 \pm 0.9\ 10^{-6}\ \mathrm{m^2\,s^{-1}}$ and $\theta_c = 0.25 \pm 0.02$ for our silica grains, and $q_0 = 2.4 \pm 0.2\ 10^{-5}\ \mathrm{m^2\,s^{-1}}$ and $\theta_c = 0.19 \pm 0.008$ for our coal grains.

These value are measured in a laminar flow, whereas our laboratory fans are produced by (mostly) turbulent channels (Sect. 2). However, regardless of the nature of the shear-inducing flow, the grain Reynolds number $d_s^2\dot\gamma/\nu$ is constant near the threshold for sediment transport ($\dot\gamma$ the vertical shear rate) (Andreotti et al., 2012). Accordingly, we use the above measurements 15 to estimate the threshold slope with equation (11).

## Appendix B: Notations

*Acknowledgements.* We thank B. Erickson, E. Steen and C. Ellis for their help in building the experimental set-up; S. Harrington and K. François-King for assistance with experiments; J-L. Grimaud for the data on sediments; L. Guerit and E. Gayer for useful discussions.

Partial financial support was provided by US National Science Foundation grants 1242458 and 1246761. P.D. work at SAFL was funded 20 by the grant of the Step'up doctoral school of IPGP and O.D. was funded by the *Émergence(s)* program of the *Mairie de Paris*, France.

**Table 4.** Variables used.

| Symbol | Definition | Unit |
|--------|------------|------|
| $Q_s$ | sediment discharge | L min$^{-1}$ |
| $Q_{s,m}$ | mass sediment discharge | g min$^{-1}$ |
| $Q_w$ | water discharge | L min$^{-1}$ |
| $\rho_s$ | sediment density | kg m$^{-3}$ |
| $\rho$ | water density | kg m$^{-3}$ |
| $g$ | acceleration of gravity | m s$^{-2}$ |
| $d_{50}, d_{90}$ | $50^{th}$ and $90^{th}$ percentile | $\mu$m |
| $\theta$ | Shield number | |
| $\theta_c$ | critical Shield number | |
| $\mu$ | friction coefficient | |
| $\tau$ | shear stress | kg m$^{-1}$ s$^{-2}$ |
| $\tau_c$ | critical shear stress | kg m$^{-1}$ s$^{-2}$ |
| $q_0$ | characteristic sediment flux | m$^2$ s$^{-1}$ |
| $W$ | width of the channel | cm |
| $\lambda$ | packing fraction | |
| $r$ | distance to the apex | m |
| $h$ | elevation | m |
| $V_c$ | volume of coal | m$^3$ |
| $V_s$ | volume of silica | m$^3$ |
| $\phi$ | proportion of silica | |
| $R_c$ | radius of coal | m |
| $R_s$ | radius of silica | m |
| $\mathcal{R}$ | radius ratio | |
| $H_c$ | elevation of the transition | m |
| $H_s$ | elevation of the fan apex | m |
| $S_c$ | distal slope | |
| $S_s$ | proximal slope | |
| $S_t$ | slope of the transition | |
| $\mathcal{S}$ | ratio of proximal to distal slope | |
| $\mathcal{S}_t$ | ratio of transition to proximal slope | |
| $S_H$ | threshold slope | |
| $n_c$ | number of channel | |
| $C_f$ | Chézy's coefficient | |
| $\mathcal{K}(1/2)$ | elliptic integral of the first kind | |

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
