# Peer review of "Self-similar growth of a bimodal laboratory fan"

_Earth Surface Dynamics, 2016_

## Referee Comment (RC1) · Anonymous Referee #1 · 23 Dec 2016

I think this is a good paper and would recommend it for publication with minor revisions. The manuscript could really use more background on field observations of alluvial fans, particularly the threshold versus transport theories of fan slope, and a discussion on how well the experiment results reflect and can be applied to real-world observations. There were a number of errors in grammar and general sentence structure. I note a few of these in the technical corrections, but the paper could use a read-through and edit by one of the native English-speaking authors.

Alluvial fans often have a single main channel, rather than many radiating from the apex. The experiments of Reitz and Jerolmack (2012) behaved similarly to real fans, with multiple channels occurring only briefly during avulsions. The experiments for this study never had fewer than 4 channels. Why is this, and how are the results applicable to real alluvial fans if they differ in this regard?

Stock et al (2008) report a similar distance between the proximal and distal fan, but

that median grain size of gravel deposits remained constant for the upper 70% of the fan. Some discussion of this would be useful.

How does the 32% length for slope transition compare with real-world fans? A couple possible sources are a databases of alluvial fans: Saito and Oguchi (2005) for humid fans, and perhaps reviews by Blissenbach (1954), Anstey (1965), or Hooke (1968) for arid fans.

You make a few references to "run 2", and it gave me the impression that you only did your analyses for that single run. Assuming you mean to say that you are using run 2 for your figures as an example, I suggest adjusting the text to make this clear (if you did only do analyses for run 2, please explain why).

Was all of the material transported as bedload, or was some portion able to transport as suspended load? Did material deposit outside of the main channel? In the distal sections (the coal only section) of the fan was flow channelized? A shift from dominantly channelized flow to dominantly overbank flow downstream might affect your assessment of fan slope being controlled by the sediment grain size. Reitz and Jerolmack (2012) report extensive overbank flow during avulsions on their experimental fans, and similar behavior has been noted for fans based on field observations (e.g. Field 2001- "Channel avulsion on alluvial fans in southern Arizona"), did your fans feature similar behavior?

In the conclusion you note that you can estimate the sediment flux that fed the fan. While this may be true for your experimental fans, the grain size distribution and flux of sediment feeding alluvial fans is essentially never constant, so when examining alluvial fan surfaces we are only really understanding the depositional processes responsible for constructing the upper few meters of the fan. In addition, fan surfaces can be reworked, masking the formative process (de Haas et al, 2014). Some discussion of this and a description of how well your results can be applied to alluvial fans in the field would be very helpful.
The many sections of the paper seem a bit convoluted. The flow of the paper would be better sections 3-6 were merged into something like "experimental setup", "model runs", and "math analyses (or something)". As it is now the division of sections 3 and 4 (as well as 5 and 6), seem a bit arbitrary, and the lines at the end of each section offering a preview of the next section are awkward. I would also suggest adding a "Notation" section as a reference for the different variables used in your equations.

Line by line comments and some technical corrections (page.line):

1.14: I think the reference here is supposed to be "Blair and McPherson (1994)-Alluvial Fan Processes and Forms". There is a new version of this book chapter from 2009 (in book "Geomorphology of Desert Environments") (the ref list has another Blair/McPherson paper from 1994)

2.5: "Perfect cone" is only the case for purely debris flow fed fans. See Williams et al (2006) "Aspects of alluvial fan shape..." (Williams et al also report that fluvially-fed alluvial fan slope-distance profiles (e.g. your figure 6b) follow an exponential fit, rather than two distinct slopes with a transition zone)

2.6: "Possible explanations for this curvature..." adding into this sentence that you are talking about the "transport" and "threshold" theories fan slope would clarify other parts of the paper where you refer to threshold theory.

2.21: "...no clear consensus..." some more detail/background on this would be helpful

3.14-26: this paragraph was hard to follow. See a few examples below:

3.16: rephrase sentence to "When unmixed, we find that for the same shear stress $\tau$, the flux of coal grains is larger than that of silica grains (Fig. 2)"

3.19: rephrase sentence to: "The shear stress required to move large grains in the mixture is lower than it would be in a system of only large grains, because they protrude more into the fluid."

3.21 "larges grains" change to "large grains"

3.22: "...different densities..." what about different diameter grains? Does this have an effect?

4.4: Is the "impervious wall" vertical?

4.13: See comment above. Alluvial fans typically have a single channel emanating from the apex which splits further downstream. The avulsion process appears different than the experiments of Reitz and Jerolmack (2012).

4.19: "Coal is deposited on the banks": is coal ever deposited overbank?

4.26: "during run 2". Did you only examine the boundary on run 2? Or do you mean to say that Fig 3a is of run 2? If the former, why not for other runs?

4.31: Same comment as for line 26

5.19: When describing similarities between profiles, do you mean all radii for a single fan, or across fans for all experiments?

5.25: See commend above (comparing the apparent match of sediment distribution change and slope transition with observations by Stock et al (2008))

7.3: the font for the variables in equations 7 and 8 is different

7.20: The sentence phrasing and grammar in this paragraph could use some editing.

7.24: Some background on the "threshold" vs "transport" theories for alluvial fan morphology would be useful.

8.33: Do you mean to say with "sediment discharge relative to water discharge" (i.e. sediment flux)?

9.20: The Conclusions section could also use some edits to grammar and sentence structure.

9.32: References here ("...in our experiments")

9.32-33: "As a consequence, we expect that the geometry of the final deposit (location and slope of the transition and proximal and distal slopes) allows us to estimate the relative flux that built the fan." This sentence seems like a big jump. The sediment supply to alluvial fans is not constant as it was in the experiments. Perhaps you mean to say the relative flux for the most recent fan deposits?

Figures:

Table 2: replace g/min-1 for sediment discharge with Lmin-1 (___is silica fraction volume or mass___...in eqn 3 it is volume) so that it uses the same units as Qw (and is easier to visualize V/V)

Fig. 3: Would it be possible to adjust contrast on photos of the fan (e.g. fig 3) so that it is easier to discern the silica against the coal?

Fig 6/7: I suggest using the same horizontal scale for these two figs.

Fig 7: Specify which run this is from (presumably run 2?)

Figure 10: this figure could probably be merged with Fig 3.

---

## Referee Comment (RC2) · A. Piliouras (Referee) · 28 Jan 2017

Overview: The authors report on physical experiments of alluvial fans with a bimodal grain size mixture, relating the grain size transition to a slope transition and comparing their experimental results to those predicted by threshold-channel theory. They conclude that the fans in their experiments grew in a self-similar manner, such that the fans maintained a consistent geometry and their growth could be described by a simple mass balance. However, the fan slope was significantly higher than that predicted by threshold channel theory, and the authors do not really provide a convincing argument for why this might be so. In general I find the paper to be well-written and wonderfully concise, although I do think some elaboration is required on the points outlined below. I recommend this paper for publication pending the following minor revisions.

Major comments:

The introduction is somewhat lacking. First, it would be helpful to relate the concepts discussed both in the intro and the present experiments to the natural environment and studies of natural fans. Second, the experiments need to be placed in a broader context to highlight why they are significant and how they advance our knowledge of alluvial fan dynamics and/or stratigraphy. This should also be revisited in the conclusions.

Your results and conclusions would be stronger by including discussion of all experiments, not just Run 2. Some of your figures seem to have other experimental data in them, but since the paper never discusses anything other than Run 2, there is somewhat of a disconnect between the text and the figures.

The discussion needs a paragraph on limitations of the experiments, particularly in their applicability to natural systems. You state in the Appendix that you did experiments with laminar flow. What, if anything, does this imply for your ability to relate these experiments to nature? How might the dynamics, geometries, and/or stratigraphies of fans created with different flow conditions differ, if at all? I do not mean to imply that you need a full discussion of hydraulic scaling (you don't), but I think that a few sentences discussing your limitations and applicability will make non-experimentalists more receptive to your ideas.

Regarding the lack of correlation between Qs and slope that comes out of the threshold channel theory analysis. Could this be because the flow is not always channelized and the deposit is not entirely formed by channels? You state in the first paragraph of section 6 that the deposit should have the same slope as the channels, but I'm not sure that this is true. I would guess that many alluvial fan and fan delta experiments, particularly thinking about those of Reitz and Jerolmack, are built by a combination of channelized and overbank or sheet flows. In that case, the overall deposit slope does not necessarily reflect the slope of a channelized flow, but perhaps that of some combination of processes. If your fan is partially formed by sheet flow or overbank deposits and not entirely formed by channels, which I suspect is likely true, then can you comment on the applicability of threshold channel theory in trying to describe a deposit that

is not and should not be the same as the bed of a channel? This may explain why the slope of the fan is quite a bit higher than the predicted threshold channel slope.

Minor comments:

Line 11: Consider adding "alluvial fans can be easily produced and boundary conditions can be easily controlled."

Line 14: Consider providing some examples of "the deposit responds by adjusting its morphology."

Line 25: Replace "At variance" with "In contrast"

Lines 25-27: You first state that Guerit et al., 2014 proposes that Qw, Qs, and grain size act independently to influence slope, but then claim that they conclude that slope depends on Qw and grain size. These sentences are in conflict and the language either needs to be adjusted to resolve it or you need to better explain the results of their study and in what ways, specifically, Qw and grain size influence slope.

Line 29: Omit the comma after "moderately"

Line 10: Include more references of experimental alluvial fans

Line 18: Omit erroneous s in "each type of grain"

Lines 19-20: "The shear stress required to move large grains in a mixture is lower than it would be in a system of uniform large grains because the grains protrude more into the fluid."

Line 21: "a higher shear stress in a bimodal mixture because they are partially shielded from the flow by neighboring large grains"

Line 27: "Below a critical shear stress"

[Figure]

Line 3: Reference your experimental setup figure here. Also consider, rearranging Section 2 to start with this information regarding your setup and procedure. This will allow you to start broader and then narrow down to the details, which will be make it read a bit more easily.

Lines 4-5: Rearrange these clauses/sentences to this order: "At the back of the tank, ... which the fan leans. At the wall's foot, ... concentrating along it. The three other sides ... evacuate water."

Line 6: Does the standing water, however shallow, influence anything about the fan's growth or toe geometry?

Line 7: Is your header tank a constant head tank? If so, say so.

Line 11: "reaches its bottom" is vague. What is "its" referring to here? The tank? Where is the bottom? Rephrase.

Line 13: Why did your experiments have five or six channels at a time? This seems in contrast to many other fan experiments, particularly with those of Reitz and Jerolmack. What are the possible causes and implications of this?

Line 19: Mustn't you also have silica deposited overbank to make the deposit shape depicted, or is silica really that narrowly deposited in the thalweg? In that case, if silica exists over much of the proximal deposit, then do the channels migrate to visit almost every point on the proximal fan in order to get that distribution or silica?

Line 24: I suggest changing the language of "eye-averaging," as it does not make your observation convincing.

Line 28: You need an extra sentence or two here to explain your image processing methods, particularly in your rescaling/stretching and error/accuracy estimates.

Line 31: Why are you only reporting on Run 2?

Line 33: Again, your measurements of accuracy are unclear. Is the 19% a standard deviation? A variance? Some roughness measurement?

Line 15: Reword to state that either your precision is better than 1mm or your error is less than 1mm.

Line 17: "This property suggests that we can compute"

Line 18: "profile of the fan with minimal error" or "with minimal loss of information"

Line 22: "We find that the slope plateaus to a value of about 0.29 near the apex and to about 0.10 near the toe."

Line 23: "transition between these slopes is smooth,"

Lines 23-24: Why are you calling this a "characteristic length?" You are only examining one experiment, or does this hold for more experiments? Please clarify.

Line 24: "55% of the fan length from the apex"

Line 26: Here you restate R $\sim$ 0.62, which closely coincides with 0.55. This would be even more convincing by stating R with the error you already have R = 0.62 +/- 0.04. Also, do you have an error on the inflection point distance 0.55? Should be stated here, if so.

Line 28: Replace "cohesive" with "intact" to avoid confusion surrounding cohesive sediment.

Line 30: "whereas coal concentrates at the fan toe."

Line 31: Replace "smeared" with "irregular" or "gradual" or "fluctuating" etc.

Line 31: "It" is vague. Rephrase to "The transitional zone shows alternating layers"

Line 4: Insert equals signs for slope: (slope = 0.29), (slope = 0.10)

Lines 5-6: "Finally we define the transition line, which joins this intersection to the origin and passes through the alternating stratigraphic layers in the transition zone."

Lines 6-7: "more mobile sediment (coal) lying below the less mobile one (silica).

Line 7: Replace "steady climb" with "upward migration"

Equation 3: Define phi in text

Line 11: How and why do you "adjust" the proximal and distal slopes?

Line 13: Your calculated silica fraction in the deposit matches that put in during experiments. Do you account for porosity in the deposit since your input flux is likely just a mass or solids volume flux? The porosities of the coal and silica are likely different, and I would expect this to influence the overall deposit volume and volume partitioning between coal and silica.

Line 21: Replace "type of sediment they flow onto" with "bed sediment composition"

Line 30: Replace "ramify" with "bifurcate"

Line 31: "threshold-channel theory slope predictions."

Line 5: "cross sections per channel per measurement strip"

Line 5: "channels and their widths in each bin over the runs."

Line 6: "distance from the apex"

Line 26: "we approximate Cf with"

Equation 10: Define all variables in text

Lines 5-13: You provide a few possible explanations for departing from theory, but you need more discussion to provide a physical reasoning for why you think this is.

Line 18: This section ends fairly abruptly.

Lines 29-30: How does this straightforwardly extend to different grain size distributions?

Line 7: "both mechanisms" this is vague. Which mechanisms?

Line 8: omit comma after "deposit"

Appendix A

Does threshold channel theory hold for laminar flow? Either a brief statement of affirmation or a brief discussion on any assumptions on this front is required.

Figure 1

Assign (a) and (b) to parts of figure. On your schematic, the text says there is also a trench at the downstream or rightmost edge, but it is not depicted here.

Figure 2

(a) This graph is somewhat confusing (particularly the vertical axis), as it is not the typical way that people in our community show grain size distributions, although I acknowledge it is mathematically accurate. Consider replotting as a "percent finer than."

Figure 3

(b) Label Rc, Rs

"The 26 pictures are each 10-minutes apart."

Figure 5

"only two sample radii 5 degrees apart"

Table 3

Are these the characteristics at the end of each run? If so, say so.

Run 5 has a drastically different R than all other experiments and much higher error. Why? It is still unclear why you only discuss Run 2 in the paper.

Figure 10

Consider rephrasing measurement "bins" rather than strips. Also applies to text.

Figure 11

The number of channels appears to decrease past the transition zone, but you claim that channels do not rejoin downstream. So do they just lose definition and you cannot detect them? This needs to be clarified.

---

## Referee Comment (RC3) · Anonymous Referee #3 · 30 Jan 2017

General comment:

I recommend this article for publishing if the comments are considered. The article is well-written and it is easy to read. The first two sections of the article are quite detailed, leading to sections that lack of substantial contributions to the general discussion by themselves, as they are too brief and without much analysis. Nevertheless, the article shows good results that are worthy to publish in spite the simple analysis with some flaws.

Major comments:

The analysis is focused only in some properties of the particles used and it misses other parameters that may give an important insight to the results, i.e. repose angles. Such properties of the materials could be included in the qualitative analysis, as Reitz and Jerolmack (2012) do.

[Figure]

It is mentioned that the exposure/hiding effects are negligible because of the density difference between the particles. This is only if the sample is well-graded. There is insufficient information provided in order to neglect, or not, such effects. Also, mobility is accounted separately, for each material, so it seems irrelevant if in the final experiments are mixed, since the mobility may be affected by the other material sizes, not only by density. Even if you are able to provide evidence that it is in fact negligible, a re-writing of the paragraph could be helpful.

There is a chapter called "Mass Balance". If you look at the equations, they are all in terms of volume. What about the packing conditions of the fan? And the packing conditions of the inlet? Why would those be the same? Is there a way to quantify the void between particles? I think that at least some assumptions should be made and explained.

I found interesting the geometrical self-similarity shown in the article, but a quite more complex self-similar behavior is there. Certainly, with the results something else could be done.

A similar pattern to the one you show when cutting the fan radially, has been obtained by other authors in a 'quasi-two-dimensional' cell, e.g. Makse et al. (1997). It could be interesting to say something about that. Makse et al. (1997) obtained stratification when the large particles' angle of repose was larger than the small one's. That is verified for Fig. 7, but what about the rest? Its quite interesting that the vertical cross section is not only segregated, but stratified. Could this be found in natural fans? If so, under which conditions? Since you performed experiments with silica volume concentrations ranging from 25% to 80%, maybe stratification depends of this parameter.

The above leads me to another comment. It seems that you only analyzed experiment 2. What about the rest?

In general for a roughly 10 pages article, 6 sections is too much I think. If some sections are merged or taken as subsections it would give more significance to each section.

As it is, seems that each section has nothing much to say, e.g. sections 4 and 5.

Minor comments:

p3.line1: It it confusing the way you say that large grains are in the upper part and small ones deposit near its toe, as figure 2 shows the opposite. The system inverses the gradation?

p3.line10: Routine seems something tedious, ordinary and repetitive, that has nothing special, therefore irrelevant. Another word could be better to start the chapter.

p4.line1: Again the density. If the density difference prevails over grain size, then how is explained that mobility has nothing to do with density? If so, which difference is more relevant? Could be there an equilibrium?

p4.line13: The number of channels is different from the number reported by Reitz and Jerolmack (2012). Is there a reason?

p4.line20: Silica proportion is introduced, is it of volume or weight? If such variable is introduced, maybe you could use a formula.

p4.line24: To put explicitly eye-average, indicates subjectivity as results may change by repeating the analysis. The error by this process is considered?

p5.line5: "The observations confirm the scaling, thus..." Instead.

p5.line24: 32% and 55% of the fan length, Which fan length? Is is the average of all the experiments? Of each experiment?

p5.line34: You say that the variability of sand-coal transition in the stratigraphy is because of channel avulsion. If you follow one of the major comments, then it is not because of that.

p6.line9: The interpretation in the context of self-similar growth should be in the self-similar growth section.
**ESurfD**

Interactive
comment

p8.line26: Why Chézy and that value? I understand that it is for simplicity, but still.

p8.line27: The reference is wrong, it should be (Chow, 1959).

p9.line20: Independently you consider the stratification analysis, you should say if deposits show stratification.

p9.line21: If you make reference to your sediments in terms of mobility it is tricky as it is true all the time and it could include both silica and coal particles. Too obvious.

p9.line30: It is not clear that it is possible to extend to a continuous size distribution. You did not say anything about the sample, where they well-graded?

p10.line2-3: The discussion about the most challenging problem is not fair enough. You lack of evidence or references to sustain that.

---

## Author Comment (AC1) · 20 Mar 2017

**Answer to reviewer 1**

**March 20, 2017**

*I think this is a good paper and would recommend it for publication with minor revisions. The manuscript could really use more background on field observations of alluvial fans, particularly the threshold versus transport theories of fan slope, and a discussion on how well the experiment results reflect and can be applied to real-world observations. There were a number of errors in grammar and general sentence structure. I note a few of these in the technical corrections, but the paper could use a read-through and edit by one of the native English-speaking authors.*

*[1] Alluvial fans often have a single main channel, rather than many radiating from the apex. The experiments of Reitz and Jerolmack (2012) behaved similarly to real fans, with multiple channels occurring only briefly during avulsions. The experiments for this study never had fewer than 4 channels. Why is this, and how are the results applicable to real alluvial fans if they differ in this regard?*

Some alluvial fans display a radial distributive pattern where all channels are active at the same time (Hartley et al., 2010). We suspect that the spread into multiple channels is due to sediment discharge (Stebbings, 1963; Métivier et al., 2016). In the experiments of Reitz and Jerolmack (2012), channels are not active simultaneously but they define a radial distributive pattern of 4 to 5 channels. We clarified this point page 6, lines 19-21.

*[2] Stock et al (2008) report a similar distance between the proximal and distal fan, but that median grain size of gravel deposits remained constant for the upper 70% of the fan. Some discussion of this would be useful.*

In our experiments, the position of the transition, hence the change in slope, depends on the proportion of silica and coal in the mixture. This accords with the observations of Stock et al. (2008), on four alluvial fans of the Mojave Desert in California, where the slope and the gravel fraction decrease with distance from fan head. We included this observation in the conclusion on page 18, lines 9-10.

*[3] How does the 32% length for slope transition compare with real-world fans? A couple possible sources are a databases of alluvial fans: Saito and Oguchi (2005) for humid fans, and perhaps reviews by Blissenbach (1954), Anstey (1965), or Hooke (1968) for arid fans.*

The length of the transition is constant in all of our experiments. To our knowledge, only Miller et al. (2014), studied this transition in the field. They showed that it is proportional to the fan length. We edited the text to include this reference page 10, lines 20-21.

*[4] You make a few references to "run 2", and it gave me the impression that you only did your analyses for that single run. Assuming you mean to say that you are using run 2 for your figures as an example, I suggest adjusting the text to make this clear (if you did only do analyses for run 2, please explain why).*

We analyzed all the runs and used run 2 to illustrate our method and results. We clarified this throughout the revised manuscript.

*[5] Was all of the material transported as bedload, or was some portion able to transport as suspended load? Did material deposit outside of the main channel? In the distal sections (the coal only section) of the fan was flow channelized? A shift from dominantly channelized flow to dominantly overbank flow downstream might affect your assessment of fan slope being controlled by the sediment grain size. Reitz and Jerolmack (2012) report extensive overbank flow during avulsions on their experimental fans, and similar behavior has been noted for fans based on field observations (e.g. Field 2001 "Channel avulsion on alluvial fans in southern Arizona"), did your fans feature similar behavior?*

In our experiments, bedload is the dominant transport mode. Only a small amount of fine coal is transported as suspended load, and overbank flow occurs temporarily during avulsions. Neither process seems to control the fan morphology, although we cannot be positive about that. We mentioned this in the revised manuscript page 6, lines 22-24.

*[6] In the conclusion you note that you can estimate the sediment flux that fed the fan. While this may be true for your experimental fans, the grain size distribution and flux of sediment feeding alluvial fans is essentially never constant, so when examining alluvial fan surfaces we are only really understanding the depositional processes responsible for constructing the upper few meters of the fan. In addition, fan surfaces can be reworked, masking the formative process (de Haas et al, 2014). Some discussion of this and a description of how well your results can be applied to alluvial fans in the field would be very helpful.*

Multiple processes can alter the fan surface and add to the primary mechanisms that control its growth (de Haas et al., 2014). Our simple experiments concentrate on basic processes. Therefore we agree that our results cannot be transposed directly to natural systems. We clarified this page 18, lines 19-23.

*[7] The many sections of the paper seem a bit convoluted. The flow of the paper would be better sections 3-6 were merged into something like "experimental setup", "model runs", and "math analyses (or something)". As it is now the division of sections 3 and 4 (as well as 5 and 6), seem a bit arbitrary, and the lines at the end of each section offering a preview of the next section are awkward. I would also suggest adding a "Notation" section as a reference for the different variables used in your equations.*

We added a Notation section in the appendix. We also removed excessive sections (3 and 4 are now merged, as well as 5 and 6). We agree that the paper could follow the plan you propose (set-up, observations and theory). However, we prefer to introduce the hypotheses upon which the theory is built (radial symmetry, self-similarity, etc.) one after the other, when our observations support them. We have significantly edited the manuscript to clarify its structure. We hope this new version is easier to read.

*Line by line comments and some technical corrections (page.line):*

*1.14: I think the reference here is supposed to be "Blair and McPherson (1994)-Alluvial Fan Processes and Forms". There is a new version of this book chapter from 2009 (in book "Geomorphology of Desert Environments") (the ref list has anotherBlair/McPherson paper from 1994)*

Done.

*2.5: "Perfect cone" is only the case for purely debris flow fed fans. See Williams et al (2006) "Aspects of alluvial fan shape. . ." (Williams et al also report that fluvially-fed alluvial fan slope-distance profiles (e.g. your figure 6b) follow an exponential fit, rather than two distinct slopes with a transition zone).*

We modified the text page 2, lines 7-9, to avoid the confusion you mention.

*2.6: "Possible explanations for this curvature. . ." adding into this sentence that you are talking about the "transport" and "threshold" theories fan slope would clarify other parts of the paper where you refer to threshold theory.*

Done.

*2.21: ". . .no clear consensus. . ." some more detail/background on this would be helpful.*

We clarified this in the revised manuscript, page 3, lines 10-12.

*3.14-26: this paragraph was hard to follow. See a few examples below:*

*3.16: rephrase sentence to "When unmixed, we find that for the same shear stress $\tau$, the flux of coal grains is larger than that of silica grains (Fig.2)"*

Done.

*3.19: rephrase sentence to: "The shear stress required to move large grains in the mixture is lower than it would be in a system of only large grains, because they protrude more into the fluid."*

Done.

*3.21 "larges grains" change to "large grains".*

Done.

*3.22: ". . .different densities. . ." what about different diameter grains? Does this have an effect?*

Both grain size and density affect the mobility of our sediment. We measured the critical Shields parameter, and found a higher value for silica (small grains, large density), than for coal (large grains, low density). This indicates that the density contrast exerts a primary influence on mobility. This is confirmed by experimental observations (page 5, lines 13-15, and page 6, lines 1-6).

*4.4: Is the "impervious wall" vertical?*

Yes it is, we added it in the revised manuscript.

*4.13: See comment above. Alluvial fans typically have a single channel emanating from the apex which splits further downstream. The avulsion process appears different than the experiments of Reitz and Jerolmack (2012).*

Some alluvial fans display a radial distributive pattern where all channels are active at the same time (Hartley et al., 2010). In the experiments of Reitz and Jerolmack (2012), channels are not active simultaneously but they define a radial distributive pattern of 4 to 5 channels. We clarified this point page 6, lines 19-21.

*4.19: "Coal is deposited on the banks": is coal ever deposited overbank?*

Overbank deposits are relatively rare, and the sediment is deposited mostly in the thalweg or on the banks. We clarified this point in the manuscript (page 6, lines 22-24).

*4.26: "during run 2". Did you only examine the boundary on run 2? Or do you mean to say that Fig 3a is of run 2? If the former, why not for other runs? 4.31: Same comment as for line 26.*

We analyzed all the runs and used run 2 to illustrate our method and results. We clarified this throughout the revised manuscript.

*5.19: When describing similarities between profiles, do you mean all radii for a single fan, or across fans for all experiments?*

We mean all radii for a single fan. We have clarified this sentence in the manuscript (page 10, line 9).

*5.25: See commend above (comparing the apparent match of sediment distribution change and slope transition with observations by Stock et al (2008))*

See our answer to your second comment.

*7.3: the font for the variables in equations 7 and 8 is different*

We checked the font of the variables.

*7.20: The sentence phrasing and grammar in this paragraph could use some editing.*

We did our best to improve the English of the manuscript.

*7.24: Some background on the "threshold" vs "transport" theories for alluvial fan morphology would be useful.*

We added a discussion on the threshold theory page 13, lines 20-22, and page 14, lines 1-2, in order to provide this background.

*8.33: Do you mean to say with "sediment discharge relative to water discharge" (i.e. sediment flux)?*

Both are true for a given water discharge, we have clarified this point in the manuscript (page 16, line 20).

*9.20: The Conclusions section could also use some edits to grammar and sentence structure.*

We edited the conclusion of the revised manuscript.

*9.32: References here ("...in our experiments")*

Done

*9.32-33: "As a consequence, we expect that the geometry of the final deposit (location and slope of the transition and proximal and distal slopes) allows us to estimate the relative flux that built the fan." This sentence seems like a big jump. The sediment supply to alluvial fans is not constant as it was in the experiments. Perhaps you mean to say the relative flux for the most recent fan deposits?*

We agree. We have changed the text in the conclusion (page 18, lines 4-11).

*Figures:*

*Table 2: replace $g/min - 1$ for sediment discharge with $Lmin - 1$ (is silica fraction volume or mass...in eqn 3 it is volume) so that it uses the same units as $Qw$ (and is easier to visualize V/V).*

Done.

*Fig. 3: Would it be possible to adjust contrast on photos of the fan (e.g. fig 3) so that it is easier to discern the silica against the coal?*

Unfortunately, the quality of the pictures does not allow us to improve the contrast much.

*Fig 6/7: I suggest using the same horizontal scale for these two figs.*

We used Fig 7 to introduce the rescaling with $R_c$. In the revised version, we have added the physical scale accords to your suggestion.

*Fig 7: Specify which run this is from (presumably run 2?)*

Done.

*Figure 10: this figure could probably be merged with Fig 3.*

We have tried merging them, but this obscured the resulting figure.

**References**

Blair, T. C.: Sedimentary processes, vertical stratification sequences, and geomorphology of the Roaring River alluvial fan, Rocky Mountain National Park, Colorado, Journal of Sedimentary Research, 57, 1987.

Blair, T. C. and McPherson, J. G.: Processes and forms of alluvial fans, in: Geomorphology of Desert Environments, pp. 413–467, Springer, 2009.

Blissenbach, E.: Relation of surface angle distribution to particle size distribution on alluvial fans, Journal of Sedimentary Research, 22, 25–28, 1952.

Bull, W. B.: Geomorphology of segmented alluvial fans in western Fresno County, California, US Government Printing Office, 1964.

de Haas, T., Ventra, D., Carbonneau, P. E., and Kleinhans, M. G.: Debris-flow dominance of alluvial fans masked by runoff reworking and weathering, Geomorphology, 217, 165–181, 2014.

Drew, F.: Alluvial and lacustrine deposits and glacial records of the Upper-Indus Basin, Quarterly Journal of the Geological Society, 29, 441–471, 1873.

Hartley, A. J., Weissmann, G. S., Nichols, G. J., and Warwick, G. L.: Large distributive fluvial systems: characteristics, distribution, and controls on development, Journal of Sedimentary Research, 80, 167–183, 2010.

Le Hooke, R. B. and Rohrer, W. L.: Geometry of alluvial fans: Effect of discharge and sediment size, Earth Surface Processes, 4, 147–166, 1979.

Métivier, F., Lajeunesse, E., and Devauchelle, O.: Laboratory rivers: Lacey's law, threshold theory and channel stability, Submitted to Earth Surface Dynamics, 2016.

Milana, J. P. and Ruzycki, L.: Alluvial-fan slope as a function of sediment transport efficiency, Journal of Sedimentary Research, 69, 1999.

Miller, K. L., Reitz, M. D., and Jerolmack, D. J.: Generalized sorting profile of alluvial fans, Geophysical Research Letters, 41, 7191–7199, 2014.

Reitz, M. D. and Jerolmack, D. J.: Experimental alluvial fan evolution: Channel dynamics, slope controls, and shoreline growth, Journal of Geophysical Research: Earth Surface (2003–2012), 117, 2012.

Saito, K. and Oguchi, T.: Slope of alluvial fans in humid regions of Japan, Taiwan and the Philippines, Geomorphology, 70, 147–162, 2005.

Stebbings, J.: The shapes of self-formed model alluvial channels., Proceedings of the Institution of Civil Engineers, 25, 485–510, 1963.

Stock, J. D., Schmidt, K. M., and Miller, D. M.: Controls on alluvial fan long-profiles, Geological Society of America Bulletin, 120, 619–640, 2008.

Weissmann, G., Bennett, G., and Lansdale, A.: Factors controlling sequence development on Quaternary fluvial fans, San Joaquin Basin, California, USA, Special Publication-Geological Society of London, 251, 169, 2005.

Weissmann, G. S., Mount, J. F., and Fogg, G. E.: Glacially driven cycles in accumulation space and sequence stratigraphy of a stream-dominated alluvial fan, San Joaquin Valley, California, USA, Journal of Sedimentary Research, 72, 240–251, 2002.

Williams, R. M., Zimbelman, J. R., and Johnston, A. K.: Aspects of alluvial fan shape indicative of formation process: A case study in southwestern California with application to Mojave Crater fans on Mars, Geophysical research letters, 33, 2006.

---

## Author Comment (AC3) · 20 Mar 2017

**Answer to reviewer 3**

March 20, 2017

*General comment*:

*I recommend this article for publishing if the comments are considered. The article is well-written and it is easy to read. The first two sections of the article are quite detailed, leading to sections that lack of substantial contributions to the general discussion by themselves, as they are too brief and without much analysis. Nevertheless, the article shows good results that are worthy to publish in spite the simple analysis with some flaws.*

*Major comments:*

*The analysis is focused only in some properties of the particles used and it misses other parameters that may give an important insight to the results, i.e. repose angles. Such properties of the materials could be included in the qualitative analysis, as Reitz and Jerolmack (2012) do.*

We quantified the repose angle of our grains. It is presented in Table 1 in term of the friction coefficient, $\mu$ which represent the tangent of this angle. This friction coefficient appears in the definition of the threshold slope (eq. 11 ). We clarified it in the revised manuscript (caption of Table 1).

*It is mentioned that the exposure/hiding effects are negligible because of the density difference between the particles. This is only if the sample is well-graded. There is insufficient information provided in order to neglect, or not, such effects. Also, mobility is accounted separately, for each material, so it seems irrelevant if in the final experiments are mixed, since the mobility may be affected by the other material sizes, not only by density. Even if you are able to provide evidence that it is in fact negligible, a re-writing of the paragraph could be helpful.*

We agree, there exist no universal transport law accounting for the hiding/exposure effect. However, to estimate, at least qualitatively, the differential mobility of our grain we use transport laws of each species and neglect the exposure and hiding effect. We re-wrote the paragraph (page 6, lines 9-17).

*There is a chapter called "Mass Balance". If you look at the equations, they are all in terms of volume. What about the packing conditions of the fan? And the packing conditions of the inlet? Why would those be the same? Is there a way to quantify the void between particles? I think that at least some assumptions should be made and explained.*

Following your comment, we measured the porosity of our granular materials, and estimated the porosity of the deposit. We found that the porosity is constant, and, therefore, does not impact the mass balance. We added a paragraph and figure 5, to explain this in the revised manuscript (page 8, lines 15-29).

*I found interesting the geometrical self-similarity shown in the article, but a quite more complex self-similar behavior is there. Certainly, with the results something else could be done.*

We agree that this preliminary work only focuses on the first-order geometry of the deposit. To further the analysis of this self-similarity, we need to understand the physical origin of the fan's slope. For this, we need to improve the geometry of the feeding channel. An experiment is under way in our laboratory. We clarified this in the conclusion, page 18, lines 19-21.

*A similar pattern to the one you show when cutting the fan radially, has been obtained by other authors in a 'quasi-two-dimensional' cell, e.g. Makse et al. (1997). It could be interesting to say something about that.*

*Makse et al. (1997) obtained stratification when the large particles' angle of repose was larger than the small one's. That is verified for Fig. 7, but what about the rest? Its quite interesting that the vertical cross section is not only segregated, but stratified. Could this be found in natural fans? If so, under which conditions? Since you performed experiments with silica volume concentrations ranging from 25% to 80%, maybe stratification depends of this parameter.*

The cross section of figure 7 shows segregated deposits separated by a mixing zone with alternating layers of silica and coal. The extent of this mixing zone (30 % of the fan length) seems to be independent of the composition of the sediment mixture. Similar stratigraphic patterns are observed in natural fan deposits, and they are interpreted as a consequence of fluctuating water and sediment discharges (Paola et al., 1992; Clevis et al., 2003; Charreau et al., 2009; Whittaker et al., 2011; Dubille and Lavé, 2015). We were not aware of the experiments and model of Makse et al. (1997b,a), that indeed exhibit a similar pattern. We refer to them in the revised manuscript (page 10, lines 7-11).

*The above leads me to another comment. It seems that you only analyzed experiment 2. What about the rest?*

We analyzed all the runs and used run 2 to illustrate our method and results. We clarified this throughout the revised manuscript.

*In general for a roughly 10 pages article, 6 sections is too much I think. If some sections are merged or taken as subsections it would give more significance to each section. As it is, seems that each section has nothing much to say, e.g. sections 4 and 5.*

Following your comment, we merged sections 3 and 4, and sections 5 and 6.

*Minor comments:*

*p3.line1: It it confusing the way you say that large grains are in the upper part and small ones deposit near its toe, as figure 2 shows the opposite. The system inverses the gradation?*

The sentence you refer to describes the experiment of Reitz and Jerolmack (2012). In their experiments the mobility difference is driven by different grain size. In our study the mobility difference is controlled essentially by the density contrast. We clarified this in the revised manuscript (page 5, lines 14-15, and page 6, lines 1-6).

*p3.line10: Routine seems something tedious, ordinary and repetitive, that has nothing special, therefore irrelevant. Another word could be better to start the chapter.*

We changed routine to common.

*p4.line1: Again the density. If the density difference prevails over grain size, then how is explained that mobility has nothing to do with density? If so, which difference is more relevant? Could be there an equilibrium?*

We quantify the grain mobility in term of the critical Shields parameter. This parameter depends on shear stress, grain size, and density. Coal grains are larger and lighter than the silica grains. The value of the critical Shield parameter is lower for coal, which suggests that the density difference prevails over the grain size difference. Experimental observations confirm this (page 5, lines 14-15, and page 6, lines 1-6).

*p4.line13: The number of channels is different from the number reported by Reitz and Jerolmack (2012). Is there a reason?*

We suspect that, in our experiment, the presence of multiple channels is due to the sediment discharge (Stebbings, 1963; Métivier et al., 2016). In the experiments of Reitz and Jerolmack (2012), channels are not active simultaneously but they define a comparable, radial, distributive pattern of 4 to 5 channels. We clarified this point page 6, lines 20-22.

*p4.line20: Silica proportion is introduced, is it of volume or weight? If such variable is introduced, maybe you could use a formula.*

Silica proportion is introduced by volume. Following your comment, we added formulas to clarify this (page 8, lines 12-22).

*p4.line24: To put explicitly eye-average, indicates subjectivity as results may change by repeating the analysis. The error by this process is considered?*

We modified the sentence to avoid misinterpretation (page 7, line 4, and page 8, line 1).

*p5.line5: "The observations confirm the scaling, thus..." Instead.*

Done.

*p5.line24: 32% and 55% of the fan length, Which fan length? Is is the average of all the experiments? Of each experiment?*

We mean the percentage of the total fan length, from apex to toe. These values are for run 2. Values for the other runs are presented in Table 3. The value of the transition length is similar for all runs. We clarified the manuscript to be more specific about this (page 10, lines 17-21).

*p5.line34: You say that the variability of sand-coal transition in the stratigraphy is because of channel avulsion. If you follow one of the major comments, then it is not because of that.*

There are two main differences between our experiments and the experiments of Makse et al. (1997b). First, the slope of the deposit is much smaller than the angle of repose due to fluid entrainment. Second our experiment is three-dimensional whereas the experiment of Makse et al. (1997b) is two-dimensional. Furthermore figure 3 shows that the limit between the two fans is strongly dissected. Because sediments are deposited mostly within channels, we believe this pattern is the result of channel avulsion. We added a discussion in the revised manuscript page 11, lines 7-11.

*p6.line9: The interpretation in the context of self-similar growth should be in the self-similar growth section.*

Following your comment, we restructured our manuscript.

*p8.line26: Why Chézy and that value? I understand that it is for simplicity, but still.*

The Chézy coefficient is the simplest possible coefficient use in fluvial experiments to quantify fluid friction (Métivier et al., 2016). Its value depends on the channel shape and grain size. We chose the value of $C_f$ according to Chow (1959).

*p8.line27: The reference is wrong, it should be (Chow, 1959).*

We corrected it.

*p9.line20: Independently you consider the stratification analysis, you should say if deposits show stratification.*

We added a sentence about stratification in the revised manuscript (page 17, lines 16-17).

*p9.line21: If you make reference to your sediments in terms of mobility it is tricky as it is true all the time and it could include both silica and coal particles. Too obvious.*

We clarified this in the revised paper.

*p9.line30: It is not clear that it is possible to extend to a continuous size distribution. You did not say anything about the sample, where they well-graded?*

We do not know whether the segregation mechanism could be extend to a continuous size distribution. But field observations are encouraging: some have shown a strong correlation between changes in slope and grain

size or sand fraction (Bull, 1964; Blair, 1987; Blair and McPherson, 2009; Miller et al., 2014; Stock et al., 2008). As a consequence, although this should be tested experimentally, we expect that the segregation occurs similarly for a continuous grain size distribution. We mention this in the revised manuscript (page 18, lines 4-11). Our sample are not well-graded, but they have distinct mobilities.

*p10.line2-3: The discussion about the most challenging problem is not fair enough. You lack of evidence or references to sustain that.*

We re-wrote the conclusion to clarify.

**References**

Blair, T. C.: Sedimentary processes, vertical stratification sequences, and geomorphology of the Roaring River alluvial fan, Rocky Mountain National Park, Colorado, Journal of Sedimentary Research, 57, 1987.

Blair, T. C. and McPherson, J. G.: Processes and forms of alluvial fans, in: Geomorphology of Desert Environments, pp. 413–467, Springer, 2009.

Bull, W. B.: Geomorphology of segmented alluvial fans in western Fresno County, California, US Government Printing Office, 1964.

Charreau, J., Gumiaux, C., Avouac, J.-P., Augier, R., Chen, Y., Barrier, L., Gilder, S., Dominguez, S., Charles, N., and Wang, Q.: The Neogene Xiyu Formation, a diachronous prograding gravel wedge at front of the Tianshan: Climatic and tectonic implications, Earth and Planetary Science Letters, 287, 298–310, 2009.

Chow, V. T.: Open channel hydraulics, 1959.

Clevis, Q., de Boer, P., and Wachter, M.: Numerical modelling of drainage basin evolution and three-dimensional alluvial fan stratigraphy, Sedimentary Geology, 163, 85–110, 2003.

Dubille, M. and Lavé, J.: Rapid grain size coarsening at sandstone/conglomerate transition: similar expression in Himalayan modern rivers and Pliocene molasse deposits, Basin Research, 27, 26–42, 2015.

Guerit, L., Métivier, F., Devauchelle, O., Lajeunesse, E., and Barrier, L.: Laboratory alluvial fans in one dimension, Physical Review E, 90, 022 203, 2014.

Le Hooke, R. B. and Rohrer, W. L.: Geometry of alluvial fans: Effect of discharge and sediment size, Earth Surface Processes, 4, 147–166, 1979.

Makse, H. A., Cizeau, P., and Stanley, H. E.: Possible stratification mechanism in granular mixtures, Physical review letters, 78, 3298, 1997a.

Makse, H. A., Havlin, S., King, P. R., and Stanley, H. E.: Spontaneous stratification in granular mixtures, Nature, 386, 379, 1997b.

Métivier, F., Lajeunesse, E., and Devauchelle, O.: Laboratory rivers: Lacey's law, threshold theory and channel stability, Submitted to Earth Surface Dynamics, 2016.

Miller, K. L., Reitz, M. D., and Jerolmack, D. J.: Generalized sorting profile of alluvial fans, Geophysical Research Letters, 41, 7191–7199, 2014.

Paola, C., Heller, P. L., and Angevine, C. L.: The large-scale dynamics of grain-size variation in alluvial basins, 1: Theory, Basin Research, 4, 73–90, 1992.

Powell, E. J., Kim, W., and Muto, T.: Varying discharge controls on timescales of autogenic storage and release processes in fluvio-deltaic environments: Tank experiments, Journal of Geophysical Research: Earth Surface (2003–2012), 117, (F2), 2012.

Reitz, M. D. and Jerolmack, D. J.: Experimental alluvial fan evolution: Channel dynamics, slope controls, and shoreline growth, Journal of Geophysical Research: Earth Surface (2003–2012), 117, 2012.

Stebbings, J.: The shapes of self-formed model alluvial channels., Proceedings of the Institution of Civil Engineers, 25, 485–510, 1963.

Stock, J. D., Schmidt, K. M., and Miller, D. M.: Controls on alluvial fan long-profiles, Geological Society of America Bulletin, 120, 619–640, 2008.

Van Dijk, M., Postma, G., and Kleinhans, M. G.: Autocyclic behaviour of fan deltas: an analogue experimental study, Sedimentology, 56, 1569–1589, 2009.

Whipple, K. X., Parker, G., Paola, C., and Mohrig, D.: Channel dynamics, sediment transport, and the slope of alluvial fans: Experimental study, The Journal of geology, 106, 677–694, 1998.

Whittaker, A. C., Duller, R. A., Springett, J., Smithells, R. A., Whitchurch, A. L., and Allen, P. A.: Decoding downstream trends in stratigraphic grain size as a function of tectonic subsidence and sediment supply, Geological Society of America Bulletin, 123, 1363–1382, 2011.

---

## Author Response (AR1)

P. Delorme
IPGP

March 20, 2017

Pr. Patricia Wiberg,
Associate Editor of Esurf

Dear Editor,

We have now completed the revision of our manuscript and responded to the three reviews provided. Enclosed are the salient points raised by the reviewers and our general answers. A detailed response is provided to each review separately.

1. All reviewers point to the interest of our bimodal fan experiment.

2. All reviewers agree with our geometrical self-similar model of fan growth.

3. All reviewers agree with our main conclusions.

4. All reviewers raise concern on the use of all the experiments, or only one run (run 2), in our analysis: this misunderstanding has now been waived and we specify more clearly that all the runs performed were analyzed and used in our calculations. Run 2 was chosen only as an example, for the figures namely.

5. All reviewers ask us to be more specific with regard to the sediment transport mode and its consequences on the fan morphology. We now specify that the sediment appears to be mainly transported as bedload, and that the influence of overbank flow seems negligible, at first order.

6. R2 (Piliouras) and R3 ask us to clarify the importance of porosity in our mass balance: we have measured the porosity of our sediment and added a figure (5). We show that this porosity is essentially a constant and does not influence our mass balance.

7. R1 and R2 ask us to develop the comparison between our experimental results and natural fans. We now develop comparisons between experimental and natural fans both in the introduction and the conclusion, and wherever possible in the manuscript. We finally acknowledge that further experiments are needed in order for this first-order model to be applicable to natural fan systems.

8. Finally R3 asks us to compare our experiments with 1D experiments on granular segregation. We now include these references, and explain in more detail the mechanism we propose for the stratification we observe between the two imbricated fans.

We hope that you will find the revised version of our manuscript suitable for publication in Esurf.

Respectfully

The authors

**Answer to reviewer 1**

March 20, 2017

*I think this is a good paper and would recommend it for publication with minor revisions. The manuscript could really use more background on field observations of alluvial fans, particularly the threshold versus transport theories of fan slope, and a discussion on how well the experiment results reflect and can be applied to real-world observations. There were a number of errors in grammar and general sentence structure. I note a few of these in the technical corrections, but the paper could use a read-through and edit by one of the native English-speaking authors.*

*[1] Alluvial fans often have a single main channel, rather than many radiating from the apex. The experiments of Reitz and Jerolmack (2012) behaved similarly to real fans, with multiple channels occurring only briefly during avulsions. The experiments for this study never had fewer than 4 channels. Why is this, and how are the results applicable to real alluvial fans if they differ in this regard?*

Some alluvial fans display a radial distributive pattern where all channels are active at the same time (Hartley et al., 2010). We suspect that the spread into multiple channels is due to sediment discharge (Stebbings, 1963; Métivier et al., 2016). In the experiments of Reitz and Jerolmack (2012), channels are not active simultaneously but they define a radial distributive pattern of 4 to 5 channels. We clarified this point page 6, lines 19-21.

*[2] Stock et al (2008) report a similar distance between the proximal and distal fan, but that median grain size of gravel deposits remained constant for the upper 70% of the fan. Some discussion of this would be useful.*

In our experiments, the position of the transition, hence the change in slope, depends on the proportion of silica and coal in the mixture. This accords with the observations of Stock et al. (2008), on four alluvial fans of the Mojave Desert in California, where the slope and the gravel fraction decrease with distance from fan head. We included this observation in the conclusion on page 18, lines 9-10.

*[3] How does the 32% length for slope transition compare with real-world fans? A couple possible sources are a databases of alluvial fans: Saito and Oguchi (2005) for humid fans, and perhaps reviews by Blissenbach (1954), Anstey (1965), or Hooke (1968) for arid fans.*

The length of the transition is constant in all of our experiments. To our knowledge, only Miller et al. (2014), studied this transition in the field. They showed that it is proportional to the fan length. We edited the text to include this reference page 10, lines 20-21.

*[4] You make a few references to "run 2", and it gave me the impression that you only did your analyses for that single run. Assuming you mean to say that you are using run 2 for your figures as an example, I suggest adjusting the text to make this clear (if you did only do analyses for run 2, please explain why).*

We analyzed all the runs and used run 2 to illustrate our method and results. We clarified this throughout the revised manuscript.

*[5] Was all of the material transported as bedload, or was some portion able to transport as suspended load? Did material deposit outside of the main channel? In the distal sections (the coal only section) of the fan was flow channelized? A shift from dominantly channelized flow to dominantly overbank flow downstream might affect your assessment of fan slope being controlled by the sediment grain size. Reitz and Jerolmack (2012) report extensive overbank flow during avulsions on their experimental fans, and similar behavior has been noted for fans based on field observations (e.g. Field 2001 "Channel avulsion on alluvial fans in southern Arizona"), did your fans feature similar behavior?*

In our experiments, bedload is the dominant transport mode. Only a small amount of fine coal is transported as suspended load, and overbank flow occurs temporarily during avulsions. Neither process seems to control the fan morphology, although we cannot be positive about that. We mentioned this in the revised manuscript page 6, lines 22-24.

[6] *In the conclusion you note that you can estimate the sediment flux that fed the fan. While this may be true for your experimental fans, the grain size distribution and flux of sediment feeding alluvial fans is essentially never constant, so when examining alluvial fan surfaces we are only really understanding the depositional processes responsible for constructing the upper few meters of the fan. In addition, fan surfaces can be reworked, masking the formative process (de Haas et al, 2014). Some discussion of this and a description of how well your results can be applied to alluvial fans in the field would be very helpful.*

Multiple processes can alter the fan surface and add to the primary mechanisms that control its growth (de Haas et al., 2014). Our simple experiments concentrate on basic processes. Therefore we agree that our results cannot be transposed directly to natural systems. We clarified this page 18, lines 19-23.

[7] *The many sections of the paper seem a bit convoluted. The flow of the paper would be better sections 3-6 were merged into something like "experimental setup", "model runs", and "math analyses (or something)". As it is now the division of sections 3 and 4 (as well as 5 and 6), seem a bit arbitrary, and the lines at the end of each section offering a preview of the next section are awkward. I would also suggest adding a "Notation" section as a reference for the different variables used in your equations.*

We added a Notation section in the appendix. We also removed excessive sections (3 and 4 are now merged, as well as 5 and 6). We agree that the paper could follow the plan you propose (set-up, observations and theory). However, we prefer to introduce the hypotheses upon which the theory is built (radial symmetry, self-similarity, etc.) one after the other, when our observations support them. We have significantly edited the manuscript to clarify its structure. We hope this new version is easier to read.

*Line by line comments and some technical corrections (page.line):*

*1.14: I think the reference here is supposed to be "Blair and McPherson (1994)-Alluvial Fan Processes and Forms". There is a new version of this book chapter from 2009 (in book "Geomorphology of Desert Environments") (the ref list has anotherBlair/McPherson paper from 1994)*

Done.

*2.5: "Perfect cone" is only the case for purely debris flow fed fans. See Williams et al (2006) "Aspects of alluvial fan shape. . ." (Williams et al also report that fluvially-fed alluvial fan slope-distance profiles (e.g. your figure 6b) follow an exponential fit, rather than two distinct slopes with a transition zone).*

We modified the text page 2, lines 7-9, to avoid the confusion you mention.

*2.6: "Possible explanations for this curvature. . ." adding into this sentence that you are talking about the "transport" and "threshold" theories fan slope would clarify other parts of the paper where you refer to threshold theory.*

Done.

*2.21: ". . .no clear consensus. . ." some more detail/background on this would be helpful.*

We clarified this in the revised manuscript, page 3, lines 10-12.

*3.14-26: this paragraph was hard to follow. See a few examples below:*

*3.16: rephrase sentence to "When unmixed, we find that for the same shear stress $\tau$, the flux of coal grains is larger than that of silica grains (Fig.2)"*

Done.

*3.19: rephrase sentence to: "The shear stress required to move large grains in the mixture is lower than it would be in a system of only large grains, because they protrude more into the fluid."*

Done.

*3.21 "larges grains" change to "large grains".*

Done.

*3.22: ". . .different densities. . ." what about different diameter grains? Does this have an effect?*

Both grain size and density affect the mobility of our sediment. We measured the critical Shields parameter, and found a higher value for silica (small grains, large density), than for coal (large grains, low density). This indicates that the density contrast exerts a primary influence on mobility. This is confirmed by experimental observations (page 5, lines 13-15, and page 6, lines 1-6).

*4.4: Is the "impervious wall" vertical?*

Yes it is, we added it in the revised manuscript.

*4.13: See comment above. Alluvial fans typically have a single channel emanating from the apex which splits further downstream. The avulsion process appears different than the experiments of Reitz and Jerolmack (2012).*

Some alluvial fans display a radial distributive pattern where all channels are active at the same time (Hartley et al., 2010). In the experiments of Reitz and Jerolmack (2012), channels are not active simultaneously but they define a radial distributive pattern of 4 to 5 channels. We clarified this point page 6, lines 19-21.

*4.19: "Coal is deposited on the banks": is coal ever deposited overbank?*

Overbank deposits are relatively rare, and the sediment is deposited mostly in the thalweg or on the banks. We clarified this point in the manuscript (page 6, lines 22-24).

*4.26: "during run 2". Did you only examine the boundary on run 2? Or do you mean to say that Fig 3a is of run 2? If the former, why not for other runs? 4.31: Same comment as for line 26.*

We analyzed all the runs and used run 2 to illustrate our method and results. We clarified this throughout the revised manuscript.

*5.19: When describing similarities between profiles, do you mean all radii for a single fan, or across fans for all experiments?*

We mean all radii for a single fan. We have clarified this sentence in the manuscript (page 10, line 9).

*5.25: See commend above (comparing the apparent match of sediment distribution change and slope transition with observations by Stock et al (2008))*

See our answer to your second comment.

*7.3: the font for the variables in equations 7 and 8 is different*

We checked the font of the variables.

*7.20: The sentence phrasing and grammar in this paragraph could use some editing.*

We did our best to improve the English of the manuscript.

*7.24: Some background on the "threshold" vs "transport" theories for alluvial fan morphology would be useful.*

We added a discussion on the threshold theory page 13, lines 20-22, and page 14, lines 1-2, in order to provide this background.

*8.33: Do you mean to say with "sediment discharge relative to water discharge" (i.e. sediment flux)?*

Both are true for a given water discharge, we have clarified this point in the manuscript (page 16, line 20).

*9.20: The Conclusions section could also use some edits to grammar and sentence structure.*

We edited the conclusion of the revised manuscript.

*9.32: References here ("...in our experiments")*

Done

*9.32-33: "As a consequence, we expect that the geometry of the final deposit (location and slope of the transition and proximal and distal slopes) allows us to estimate the relative flux that built the fan." This sentence seems like a big jump. The sediment supply to alluvial fans is not constant as it was in the experiments. Perhaps you mean to say the relative flux for the most recent fan deposits?*

We agree. We have changed the text in the conclusion (page 18, lines 4-11).

*Figures:*

*Table 2: replace $g/min - 1$ for sediment discharge with $Lmin - 1$ (is silica fraction volume or mass...in eqn 3 it is volume) so that it uses the same units as $Qw$ (and is easier to visualize V/V).*

Done.

*Fig. 3: Would it be possible to adjust contrast on photos of the fan (e.g. fig 3) so that it is easier to discern the silica against the coal?*

Unfortunately, the quality of the pictures does not allow us to improve the contrast much.

*Fig 6/7: I suggest using the same horizontal scale for these two figs.*

We used Fig 7 to introduce the rescaling with $R_c$. In the revised version, we have added the physical scale accords to your suggestion.

*Fig 7: Specify which run this is from (presumably run 2?)*

Done.

*Figure 10: this figure could probably be merged with Fig 3.*

We have tried merging them, but this obscured the resulting figure.

**Answer to A. Piliouras**

March 20, 2017

*Overview: The authors report on physical experiments of alluvial fans with a bimodal grain size mixture, relating the grain size transition to a slope transition and comparing their experimental results to those predicted by threshold-channel theory. They conclude that the fans in their experiments grew in a self-similar manner, such that the fans maintained a consistent geometry and their growth could be described by a simple mass balance. However, the fan slope was significantly higher than that predicted by threshold channel theory, and the authors do not really provide a convincing argument for why this might be so. In general I find the paper to be well-written and wonderfully concise, although I do think some elaboration is required on the points outlined below. I recommend this paper for publication pending the following minor revisions.*

*Major comments:*

*The introduction is somewhat lacking. First, it would be helpful to relate the concepts discussed both in the intro and the present experiments to the natural environment and studies of natural fans. Second, the experiments need to be placed in a broader context to highlight why they are significant and how they advance our knowledge of alluvial fan dynamics and/or stratigraphy. This should also be revisited in the conclusions.*

We added more references about fan morphology in the introduction (page 2, lines 11-13, and 19-20, and page 3, lines 24-25 ). We also discussed the applicability of our experiments in the conclusion ( page 18, lines 14-23).

*Your results and conclusions would be stronger by including discussion of all experiments, not just Run 2. Some of your figures seem to have other experimental data in them, but since the paper never discusses anything other than Run 2, there is somewhat of a disconnect between the text and the figures.*

Run 2 was used to illustrate our method and results, but we analyzed all runs. We clarified this throughout the revised manuscript.

*The discussion needs a paragraph on limitations of the experiments, particularly in their applicability to natural systems. You state in the Appendix that you did experiments with laminar flow. What, if anything, does this imply for your ability to relate these experiments to nature? How might the dynamics, geometries, and/or stratigraphies of fans created with different flow conditions differ, if at all? I do not mean to imply that you need a full discussion of hydraulic scaling (you don't), but I think that a few sentences discussing your limitations and applicability will make non-experimentalists more receptive to your ideas.*

We use laminar flows only to calibrate the transport laws, and measure the critical Shields parameters of silica and coal, we clarified this in the revised manuscript (page 6, lines 24-55, and page 19, lines 17-20). Nevertheless, we acknowledge that our results do not directly apply to natural systems, and we added a paragraph on limitation of the experiments in the conclusion (page 18, lines 14-23).

*Regarding the lack of correlation between Qs and slope that comes out of the threshold channel theory analysis. Could this be because the flow is not always channelized and the deposit is not entirely formed by channels? You state in the first paragraph of section 6 that the deposit should have the same slope as the channels, but I'm not sure that this is true. I would guess that many alluvial fan and fan delta experiments, particularly thinking about those of Reitz and Jerolmack, are built by a combination of channelized and overbank or sheet flows. In that case, the overall deposit slope does not necessarily reflect the slope of a channelized flow, but perhaps that of some combination of processes. If your fan is partially formed by sheet flow or overbank deposits and not entirely formed by channels, which I suspect is likely true, then can you comment on the applicability of threshold*

*channel theory in trying to describe a deposit that is not and should not be the same as the bed of a channel? This may explain why the slope of the fan is quite a bit higher than the predicted threshold channel slope.*

When looking at our experiments, we could see the grains moving only in channels (page 6, lines 20-24). From this observation, we hypothesize that the sediment is mainly deposited within the channels or on the banks. The primary transport mode is bedload, and overbank deposition only occurs transiently during avulsions. Moreover, we do not show terraces on the DEM. Thus, although we don't know how overbank flows may affect the slope of the deposit, we suspect their effect to be minor. We clarified this page 13, lines 19-20.

*Minor comments:*

*Page 2*

*Line 11: Consider adding "alluvial fans can be easily produced and boundary conditions can be easily controlled."*

Done.

*Line 14: Consider providing some examples of "the deposit responds by adjusting its morphology."*

We added a reference to the work of Muto and Steel (2004) in the manuscript (page 2, lines 20-21) as an example of this adjustment.

*Line 25: Replace "At variance" with "In contrast".*

Done.

*Lines 25-27: You first state that Guerit et al., 2014 proposes that Qw, Qs, and grain size act independently to influence slope, but then claim that they conclude that slope depends on Qw and grain size. These sentences are in conflict and the language either needs to be adjusted to resolve it or you need to better explain the results of their study and in what ways, specifically, Qw and grain size influence slope.*

Guerit et al. (2014) used uniform sediment, and did not explore the influence of grain size but focused on the respective influence of the water and sediment discharges. They found that, for a given grain size, $Q_w$ is the first-order control on the slope, to which $Q_s$ adds only a perturbation. We clarified in the revised manuscript (page 3, lines 12-17).

*Line 29: Omit the comma after "moderately".*

Done.

*Page 3*

*Line 10: Include more references of experimental alluvial fans.*

Done.

*Line 18: Omit erroneous s in "each type of grain".*

Done.

*Lines 19-20: "The shear stress required to move large grains in a mixture is lower than it would be in a system of uniform large grains because the grains protrude more into the fluid."*

Done.

*Line 21: "a higher shear stress in a bimodal mixture because they are partially shielded from the flow by neighboring large grains".*

Done.

*Line 27: "Below a critical shear stress".*

Done.

*Page 4*

*Line 3: Reference your experimental setup figure here.*

Done.

*Also consider, rearranging Section 2 to start with this information regarding your setup and procedure. This will allow you to start broader and then narrow down to the details, which will be make it read a bit more easily.*

Done.

*Lines 4-5: Rearrange these clauses/sentences to this order: "At the back of the tank,. . . which the fan leans. At the wall's foot, . . . concentrating along it. The three other sides . . . evacuate water."*

Done.

*Line 6: Does the standing water, however shallow, influence anything about the fan's growth or toe geometry?*

Standing water induces the formation of a fan delta (Powell et al., 2012). Yet, as the immersed part of the fan does not represent more than 1% of the total fan volume we can neglect it in the mass balance, and therefore in our analysis. We clarified that in the revised manuscript page 4, lines 10-11, and page 5, lines 1-2.

*Line 7: Is your header tank a constant head tank? If so, say so.*

Yes it is a constant-head tank. We clarified the text.

*Line 11: "reaches its bottom" is vague. What is "its" referring to here? The tank? Where is the bottom? Rephrase.*

Its refers to the tank bottom. We rephrased it (page 6, lines 18-19).

*Line 13: Why did your experiments have five or six channels at a time? This seems in contrast to many other fan experiments, particularly with those of Reitz and Jerolmack. What are the possible causes and implications of this?*

Hartley et al. (2010) have shown that alluvial fans can display a radial distributive pattern where all channels are active at the same time. We suspect that the multiple channels are due to sediment discharge (Stebbings, 1963; Métivier et al., 2016). In the experiments of Reitz and Jerolmack (2012), channels are not active simultaneously but they define a radial distributive pattern of 4 to 5 channels. We clarified this point page 6, lines 20-22.

*Line 19: Mustn't you also have silica deposited overbank to make the deposit shape depicted, or is silica really that narrowly deposited in the thalweg? In that case, if silica exists over much of the proximal deposit, then do the channels migrate to visit almost every point on the proximal fan in order to get that distribution or silica?*

Looking at our experiments, we observe that silica is indeed deposited in the thalweg. Avulsion process, however, allows the channel to visit every point of the fan. We clarified this in the manuscript page 6, lines 22-24.

*Line 24: I suggest changing the language of "eye-averaging," as it does not make your observation convincing.*

Done, page 7, line 4, and page 8, line 1.

*Line 28: You need an extra sentence or two here to explain your image processing methods, particularly in your rescaling/stretching and error/accuracy estimates.*

We added an extra sentence to explain our image procedure (page 8, line 3-5).

*Line 31: Why are you only reporting on Run 2?*

We analyzed all the runs and used run 2 to illustrate our method and results. We clarified this throughout the revised manuscript.

*Line 33: Again, your measurements of accuracy are unclear. Is the 19% a standard deviation? A variance? Some roughness measurement?*

The 19% value corresponds to a standard deviation, we added this precision in the revised manuscript (page 8, line 10).

*Page 5*

*Line 15: Reword to state that either your precision is better than 1mm or your error is less than 1mm.*

Done.

*Line 17: "This property suggests that we can compute".*

Done.

*Line 18: "profile of the fan with minimal error" or "with minimal loss of information".*

Done.

*Line 22: "We find that the slope plateaus to a value of about 0.29 near the apex and to about 0.10 near the toe."*

Done.

*Line 23: "transition between these slopes is smooth,"*

Done.

*Lines 23-24: Why are you calling this a "characteristic length?" You are only examining one experiment, or does this hold for more experiments? Please clarify.*

The length of the transition is the same for all runs. We clarified it in the revised manuscript (page 10, lines 19-20).

*Line 24: "55% of the fan length from the apex".*

Done.

*Line 26: Here you restate R ≈ 0.62, which closely coincides with 0.55. This would be even more convincing by stating R with the error you already have R = 0.62 ± 0.04. Also, do you have an error on the inflection point distance 0.55? Should be stated here, if so.*

Done.

*Line 28: Replace "cohesive" with "intact" to avoid confusion surrounding cohesive sediment.*

Done.

*Line 30: "whereas coal concentrates at the fan toe."*

Done.

*Line 31: Replace "smeared" with "irregular" or "gradual" or "fluctuating" etc.*

Done.

*Line 31: "It" is vague. Rephrase to "The transitional zone shows alternating layers".*

Done

*Page 6*

*Line 4: Insert equals signs for slope: (slope = 0.29), (slope = 0.10).*

Done.

*Lines 5-6: "Finally we define the transition line, which joins this intersection to the origin and passes through the alternating stratigraphic layers in the transition zone."*

Done.

*Lines 6-7: "more mobile sediment (coal) lying below the less mobile one (silica).*

Done.

*Line 7: Replace "steady climb" with "upward migration".*

Done.

*Equation 3: Define phi in text.*

Done.

*Page 7*

*Line 11: How and why do you "adjust" the proximal and distal slopes?*

We use a linear fit to quantify the slopes of the deposit. We explained this in the revised manuscript (page 13, lines 10-11).

*Line 13: Your calculated silica fraction in the deposit matches that put in during experiments. Do you account for porosity in the deposit since your input flux is likely just a mass or solids volume flux? The porosities of the coal and silica are likely different, and I would expect this to influence the overall deposit volume and volume partitioning between coal and silica.*

Following your comment, we measured the porosity of our granular materials and estimated the porosity of the deposit. Within uncertainties, they are the same. We added a paragraph and Fig.(5) to explain these measurements in the revised manuscript (page 8, lines 15-29).

*Line 21: Replace "type of sediment they flow onto" with "bed sediment composition".*

Done.

*Line 30: Replace "ramify" with "bifurcate".*

Done.

*Line 31: "threshold-channel theory slope predictions."*

Done.

*Page 8*

*Line 5: "cross sections per channel per measurement strip".*

Done.

*Line 5: "channels and their widths in each bin over the runs."*

Done.

*Line 6: "distance from the apex".*

Done.

*Line 26: "we approximate Cf with".*

Done.

*Equation 10: Define all variables in text.*

We have carefully checked that all variables are defined in the text of our manuscript. In addition, we added a Notation section in the appendix where all variables used are defined.

*Page 9*

*Lines 5-13: You provide a few possible explanations for departing from theory, but you need more discussion to provide a physical reasoning for why you think this is.*

Guerit et al. (2014) modeled the influence of sediment discharge on the fan profile, and showed that the resulting fan slope is steeper than the threshold slope. Experiments are under way in our group to test this hypothesis. We modified the text to include this discussion page 16, lines 27-33, and page 17, lines 1-2.

*Line 18: This section ends fairly abruptly.*

We agree, we edited the revised manuscript (page 17, lines 8-10).

*Lines 29-30: How does this straightforwardly extend to different grain size distributions?*

Indeed, it might not be so straightforward. Field observations, however, are encouraging: some have shown a strong correlation between changes in slope and grain size or sand fraction (Bull, 1964; Blair, 1987; Blair and McPherson, 2009; Miller et al., 2014; Stock et al., 2008). We mentioned this in the revised manuscript (page 18, lines 4-10).

*Page 10*

*Line 7: "both mechanisms" this is vague. Which mechanisms?*

We clarified this in the revised manuscript (page 18 , lines 28-32).

*Line 8: omit comma after "deposit".*

Done.

*Appendix A*

*Does threshold channel theory hold for laminar flow? Either a brief statement of affirmation or a brief discussion on any assumptions on this front is required.*

Seizilles et al. (2013) have shown that the threshold theory indeed applies to laminar flows, but we only use laminar flows to measure the critical Shields parameter anyway. We clarified this in the revised manuscript (page 19, line 17-20).

*Figure 1 Assign (a) and (b) to parts of figure. On your schematic, the text says there is also a trench at the downstream or rightmost edge, but it is not depicted here.*

Done.

*Figure 2 (a) This graph is somewhat confusing (particularly the vertical axis), as it is not the typical way that people in our community show grain size distributions, although I acknowledge it is mathematically accurate. Consider replotting as a "percent finer than."*

Done.

*Figure 3 (b) Label Rc, Rs.*

The picture is rescaled there is no Rc and Rs but $\mathcal{R} = \dfrac{R_s}{R_c}$ and 1.

*"The 26 pictures are each 10-minutes apart."*

Done.

*Figure 5 "only two sample radii 5 degrees apart".*

Done.

*Table 3 Are these the characteristics at the end of each run? If so, say so. Run 5 has a drastically different R than all other experiments and much higher error. Why? It is still unclear why you only discuss Run 2 in the paper.*

These are characteristics at the end of each run, Run 5 has a drastically different $\mathcal{R}$ because it involves a mixture of 80% of silica. As a consequence, the silica-coal transition is more distal than in the other experiments. However, on Fig. 10, this run does not appears as an outlier. The error is due to fluctuations of the silica-coal transition. We clarified this it the revised manuscript ( Table 3).

*Figure 10 Consider rephrasing measurement "bins" rather than strips. Also applies to text.*

Done.

*Figure 11 The number of channels appears to decrease past the transition zone, but you claim that channels do not rejoin downstream. So do they just lose definition and you cannot detect them? This needs to be clarified.*

Yes, they are more difficult to detect in the distal part of the fan, due to the poor contrast. We clarified this in the manuscript (page 16, lines 4-5).

We quantified the repose angle of our grains. It is presented in Table 1 in term of the friction coefficient, $\mu$ which represent the tangent of this angle. This friction coefficient appears in the definition of the threshold slope (eq. 11 ). We clarified it in the revised manuscript (caption of Table 1).

*It is mentioned that the exposure/hiding effects are negligible because of the density difference between the particles. This is only if the sample is well-graded. There is insufficient information provided in order to neglect, or not, such effects. Also, mobility is accounted separately, for each material, so it seems irrelevant if in the final experiments are mixed, since the mobility may be affected by the other material sizes, not only by density. Even if you are able to provide evidence that it is in fact negligible, a re-writing of the paragraph could be helpful.*

We agree, there exist no universal transport law accounting for the hiding/exposure effect. However, to estimate, at least qualitatively, the differential mobility of our grain we use transport laws of each species and neglect the exposure and hiding effect. We re-wrote the paragraph (page 6, lines 9-17).

*There is a chapter called "Mass Balance". If you look at the equations, they are all in terms of volume. What about the packing conditions of the fan? And the packing conditions of the inlet? Why would those be the same? Is there a way to quantify the void between particles? I think that at least some assumptions should be made and explained.*

Following your comment, we measured the porosity of our granular materials, and estimated the porosity of the deposit. We found that the porosity is constant, and, therefore, does not impact the mass balance. We added a paragraph and figure 5, to explain this in the revised manuscript (page 8, lines 15-29).

*I found interesting the geometrical self-similarity shown in the article, but a quite more complex self-similar behavior is there. Certainly, with the results something else could be done.*

We agree that this preliminary work only focuses on the first-order geometry of the deposit. To further the analysis of this self-similarity, we need to understand the physical origin of the fan's slope. For this, we need to improve the geometry of the feeding channel. An experiment is under way in our laboratory. We clarified this in the conclusion, page 18, lines 19-21.

*A similar pattern to the one you show when cutting the fan radially, has been obtained by other authors in a 'quasi-two-dimensional' cell, e.g. Makse et al. (1997). It could be interesting to say something about that.*

*Makse et al. (1997) obtained stratification when the large particles' angle of repose was larger than the small one's. That is verified for Fig. 7, but what about the rest? Its quite interesting that the vertical cross section is not only segregated, but stratified. Could this be found in natural fans? If so, under which conditions? Since you performed experiments with silica volume concentrations ranging from 25% to 80%, maybe stratification depends of this parameter.*

The cross section of figure 7 shows segregated deposits separated by a mixing zone with alternating layers of silica and coal. The extent of this mixing zone (30 % of the fan length) seems to be independent of the composition of the sediment mixture. Similar stratigraphic patterns are observed in natural fan deposits, and they are interpreted as a consequence of fluctuating water and sediment discharges (Paola et al., 1992; Clevis et al., 2003; Charreau et al., 2009; Whittaker et al., 2011; Dubille and Lavé, 2015). We were not aware of the experiments and model of Makse et al. (1997b,a), that indeed exhibit a similar pattern. We refer to them in the revised manuscript (page 10, lines 7-11).

*The above leads me to another comment. It seems that you only analyzed experiment 2. What about the rest?*

We analyzed all the runs and used run 2 to illustrate our method and results. We clarified this throughout the revised manuscript.

*In general for a roughly 10 pages article, 6 sections is too much I think. If some sections are merged or taken as subsections it would give more significance to each section. As it is, seems that each section has nothing much to say, e.g. sections 4 and 5.*

Following your comment, we merged sections 3 and 4, and sections 5 and 6.

*Minor comments:*

*p3.line1: It it confusing the way you say that large grains are in the upper part and small ones deposit near its toe, as figure 2 shows the opposite. The system inverses the gradation?*

The sentence you refer to describes the experiment of Reitz and Jerolmack (2012). In their experiments the mobility difference is driven by different grain size. In our study the mobility difference is controlled essentially by the density contrast. We clarified this in the revised manuscript (page 5, lines 14-15, and page 6, lines 1-6).

*p3.line10: Routine seems something tedious, ordinary and repetitive, that has nothing special, therefore irrelevant. Another word could be better to start the chapter.*

We changed routine to common.

*p4.line1: Again the density. If the density difference prevails over grain size, then how is explained that mobility has nothing to do with density? If so, which difference is more relevant? Could be there an equilibrium?*

We quantify the grain mobility in term of the critical Shields parameter. This parameter depends on shear stress, grain size, and density. Coal grains are larger and lighter than the silica grains. The value of the critical Shield parameter is lower for coal, which suggests that the density difference prevails over the grain size difference. Experimental observations confirm this (page 5, lines 14-15, and page 6, lines 1-6).

*p4.line13: The number of channels is different from the number reported by Reitz and Jerolmack (2012). Is there a reason?*

We suspect that, in our experiment, the presence of multiple channels is due to the sediment discharge (Stebbings, 1963; Métivier et al., 2016). In the experiments of Reitz and Jerolmack (2012), channels are not active simultaneously but they define a comparable, radial, distributive pattern of 4 to 5 channels. We clarified this point page 6, lines 20-22.

*p4.line20: Silica proportion is introduced, is it of volume or weight? If such variable is introduced, maybe you could use a formula.*

Silica proportion is introduced by volume. Following your comment, we added formulas to clarify this (page 8, lines 12-22).

*p4.line24: To put explicitly eye-average, indicates subjectivity as results may change by repeating the analysis. The error by this process is considered?*

We modified the sentence to avoid misinterpretation (page 7, line 4, and page 8, line 1).

*p5.line5: "The observations confirm the scaling, thus..." Instead.*

Done.

*p5.line24: 32% and 55% of the fan length, Which fan length? Is is the average of all the experiments? Of each experiment?*

We mean the percentage of the total fan length, from apex to toe. These values are for run 2. Values for the other runs are presented in Table 3. The value of the transition length is similar for all runs. We clarified the manuscript to be more specific about this (page 10, lines 17-21).

*p5.line34: You say that the variability of sand-coal transition in the stratigraphy is because of channel avulsion. If you follow one of the major comments, then it is not because of that.*

There are two main differences between our experiments and the experiments of Makse et al. (1997b). First, the slope of the deposit is much smaller than the angle of repose due to fluid entrainment. Second our experiment is three-dimensional whereas the experiment of Makse et al. (1997b) is two-dimensional. Furthermore figure 3 shows that the limit between the two fans is strongly dissected. Because sediments are deposited mostly within channels, we believe this pattern is the result of channel avulsion. We added a discussion in the revised manuscript page 11, lines 7-11.

*p6.line9: The interpretation in the context of self-similar growth should be in the self-similar growth section.*

Following your comment, we restructured our manuscript.

*p8.line26: Why Chézy and that value? I understand that it is for simplicity, but still.*

The Chézy coefficient is the simplest possible coefficient use in fluvial experiments to quantify fluid friction (Métivier et al., 2016). Its value depends on the channel shape and grain size. We chose the value of $C_f$ according to Chow (1959).

*p8.line27: The reference is wrong, it should be (Chow, 1959).*

We corrected it.

*p9.line20: Independently you consider the stratification analysis, you should say if deposits show stratification.*

We added a sentence about stratification in the revised manuscript (page 17, lines 16-17).

*p9.line21: If you make reference to your sediments in terms of mobility it is tricky as it is true all the time and it could include both silica and coal particles. Too obvious.*

We clarified this in the revised paper.

*p9.line30: It is not clear that it is possible to extend to a continuous size distribution. You did not say anything about the sample, where they well-graded?*

We do not know whether the segregation mechanism could be extend to a continuous size distribution. But field observations are encouraging: some have shown a strong correlation between changes in slope and grain

size or sand fraction (Bull, 1964; Blair, 1987; Blair and McPherson, 2009; Miller et al., 2014; Stock et al., 2008). As a consequence, although this should be tested experimentally, we expect that the segregation occurs similarly for a continuous grain size distribution. We mention this in the revised manuscript (page 18, lines 4-11). Our sample are not well-graded, but they have distinct mobilities.

*p10.line2-3: The discussion about the most challenging problem is not fair enough. You lack of evidence or references to sustain that.*

We re-wrote the conclusion to clarify.

**References**

[revised manuscript text omitted]

[91]removed: during run 2.

[92]removed: precision

[93]removed: We

[94]removed: (Powell et al., 2012)

[95]removed: volume $V$ of the fan

[96]removed: the duration of the experiment:

[99]removed: observations

[100]removed: fan. Accordingly,

[Figure]

**Figure 5.** Packing fraction of the deposit as a function of the composition of the sediment mixture. Blue dots calculated from experiment. Red dots measured independently. The red dashed line is the mean packing fraction measured independently.

location of the transition, defined by the ratio $\mathcal{R} = R_s / R_c$, remains constant [..[101] ]throughout growth ($\mathcal{R} = 0.62 \pm 0.04$ for run 2, other runs are presented in Table 3, Figs.3b and 4).

5      This self-similarity means that, as it grows, the fan preserves its structure, which can therefore be extrapolated from the final deposit. [..[102] ]

**4    [..[103] ]**

A few minutes after the experiment stops, all the surface water has drained away from the fan, leaving the entire deposit emergent. At this point, we scan the deposit's surface with a laser to measure its topography (OptoEngine MRL-FN-671, 1 W,

10    671 nm). A line generator converts the beam into a laser sheet (60° opening angle, 1 mm thick), the intersection of which with the fan surface is recorded by a camera attached to the laser, about 2 m above the tank bottom (Sick Ranger E50, 12.5 mm lens). The precision of the measurement is [..[104] ]better than 1 mm in every direction.

Using the digital elevation model (DEM) of our experimental fan, we compute the final volume of our fans to check the total packing fraction of the deposit (blue dots, Fig. 5). Despite some dispersion, we find that the [..[105] ]packing fraction of our deposit is about 54% $\pm$ 2%, close to the value estimated independently.

The elevation contours of the DEM are well approximated by concentric circles, another indication of radial symmetry (Fig. 6). This property suggests that we can compute the radially-averaged profile of the fan with minimal loss of information
* * *
[101] removed: during fan

[102] removed: In the next section, we describe the final deposit, through its slope and radial cross-section.

[103] removed: Two imbricated fans

[104] removed: less

[105] removed: elevation contours

[Figure]

**Figure 6.** Digital elevation model of an experimental fan (run 2). Black lines: elevation contours 15 mm apart from each other. White dashed lines indicate the bounds used for averaging (only two sample radii 5 degree apart are represented for clarity).

[Figure]

**Figure 7.** (a) Fan profiles at different angles (run 2). Gray: individual profiles; magenta: average profile. (b) Average downstream slope (magenta). Fitted hyperbolic tangent (dashed gray). Inflection point (gray dot) and boundary of the transition area (vertical dashed gray line).

5  (Reitz and Jerolmack, 2012). To do so, we interpolate the DEM along 34 radii, 5° apart from each other, [..[106] ]at the end of each run (Fig. 6). [..[107] ]For each run, the resulting profiles are similar to each other, and differ from the mean by less than 7% (Fig. 7a). The average fan profile is steeper near the apex than at the toe and can be approximated by two segments of uniform slope. Natural fans sometimes feature a similarly segmented profile (Bull, 1964; Blair and McPherson, 2009; Miller et al., 2014). When we plot the downstream slope of this average profile as a function of the distance to the apex, the

10  transition appears as a decreasing sigmoid curve (Fig. 7b). To [..[108] ]evaluate the location of the transition, and the extension of the transition zone, we fit a hyperbolic tangent to the slope profile [..[109] ](Fig. 7b for run 2, other runs in Table 3). For run 2, we find that the slope plateaus [..[110] ]to a value of about 0.29 near the apex, and [..[111] ]to about 0.10 near the toe. [..[112] ]We define the location of the transition [..[113] ]as the inflection point of the [..[114] ]sigmoid, which occurs at 55% [..[115] ]$\pm$ 9% of the total fan length (Fig. 7b). The slope thus breaks [..[116] ]where the sediment turns to coal, suggesting that these transitions

15  are closely [..[117] ]related (Fig. 3, [..[118] ]$\mathcal{R} \approx 0.62 \pm 0.04$). The location of the transition depends on the mixture composition (Table 3). We now define the extension of the transition zone as the characteristic length of the sigmoid. For run 2, we find that the transition between the two segments of the fan occurs over a length of 32% of the total fan length. This value is almost independent of the sediment mixture (about 30% $\pm$ 3% on average for all runs). Miller et al. (2014) found a comparable value (about 22%) for natural and laboratory fans.

20  To [..[119] ]investigate the relation between the slope break and the silica-coal transition, we now turn our attention to the internal structure of the deposit. After the water and sediment supplies have been switched off, the fan remains [..[120] ]intact, and we can cut it radially to reveal a vertical cross section (Fig. 8). Silica and coal appear segregated, in accordance with the top-view pictures of the fan (Fig. 3), and with the experiments of Reitz and Jerolmack (2012). Silica concentrates near the apex, in the upper part of the deposit, whereas coal [..[121] ]concentrates at the fan toe. The [..[122] ]location of the silica-coal transition fluctuates, and generate an intricate stratigraphy that combines segregation at the fan scale, and stratification near the transition. The transition zone shows alternating layers of silica and coal, which extend over about one third of the cross-section area. [..[123] ]In natural fans, such stratifications result from fluctuations of the sediment and water discharges,
* * *
[106] removed: using the Scipy Ndimage library

[107] removed: The

[108] removed: quantify this observation

[109] removed: . We

[110] removed: at

[111] removed: at

[revised manuscript text omitted]

where we have defined the ratio of proximal slope to distal slope $\mathcal{S} = S_s/S_c$. Equivalently we may express the composition of the sediment mixture as a function of the slope ratio and the slope of the transition:

$$\phi = \frac{1 - \mathcal{S}_t}{1 + \mathcal{S}_t \left((\mathcal{S}\mathcal{S}_t)^2 + 3\mathcal{S}\mathcal{S}_t + 2\right)}, \tag{10}$$

where we have defined the ratio of transition slope to proximal slope $\mathcal{S}_t = S_t/S_s$.

If the template is a reasonable representation of the fan geometry, the location and the slope of the transition and the two surface slopes of the deposit should adjust to the composition of the sediment input, according to Eqs. (9, 10). To evaluate this model, we measure the geometry of the fan at the end of every experimental run (Table 3). Using the radially averaged profile [..[130] ]we first fit, using a linear regression, the proximal and distal slopes and calculate their ratio. Then, we estimate the location of the transition using the position of the inflection point (Sect. [..[131] ]3). We find that, for all runs, the proportion of silica in the deposit, as deduced from our measurements through Eqs. (9, 10), matches the composition of the sediment mixture (Fig. 10).

[..[132] ]

At first order, we can thus represent our experimental fan as a radially symmetric, fully segregated structure which preserves its shape as it grows. These features determine the dynamics of the fan, and the geometry of its deposit. This model, however, involves two free parameters: the proximal and distal slopes. These are selected by the fan itself, by a mechanism that remains to be understood. [..[133] ]

**5 [..[134] ]**
* * *
[130] removed: , we first adjust

[131] removed: ??

[132] removed: Despite some imperfections in the experimental set-up and a complex stratigraphy, we find that we can reasonably

[133] removed: The next section addresses this problem.

[134] removed: Fan slope

[Figure]

**Figure 10.** Proportion of silica inferred from the geometry of the deposit, after Eq. (9) (blue) and after Eq. (10) (green), as a function of the composition of the sediment input. Red line: perfect agreement.

[Figure]

**Figure 11.** Top-view of an experimental fan superimposed with measurement bins (white), and channels cross sections (blue).

15 [..[135] ]

    [..[136] ]Each deposit is built by a collection of channels, which select their own slope according to the [..[137] ]composition of the bed, and to their sediment and water discharges. [..[138] ]On the DEM of our experimental fans, the channels are virtually invisible, showing that their downstream slope is that of the fan (Fig. 6). It is thus reasonable to assume that the deposit inherits the slope of the channels that build it.

    The way a river selects its morphology is still a matter of debate, but it has been recently pointed out that most laboratory

5 rivers, including those flowing over an experimental fan, [..[139] ]remain near the threshold for sediment transport (Reitz
* * *
[135] removed: Top-view of an experimental fan superimposed with measurement strips (white), and channels cross-sections (blue).

[136] removed: Two imbricated deposits make up our experimental fan.

[137] removed: type of sediment they flow onto

[138] removed: The deposit then

[139] removed: compare well with the threshold-channel theory

[Figure]

**Figure 12.** Evolution of active channels for all the runs. Channel width as a function of the dimensionless radius (a) and time (b). Number of channels as a function of dimensionless radius (c) and time (d). Black dashed line: average. Shaded area: variability over experimental runs.

and Jerolmack, 2012; Seizilles et al., 2013; Reitz et al., 2014; Métivier et al., 2016b). [..140 ]Assuming a channel is exactly at threshold yields a theoretical relationship between its water discharge and its slope (Glover and Florey, 1951; Henderson, 1961). Could this theory inform us about the slope of our fans?

Returning to our experimental fans, we find them enmeshed in a collection of channels flowing radially (Fig. 11). These channels sometimes [..141 ]bifurcate downstream, but do not recombine as they would in a braided river. [..142 ]We would like to compare their slope to the prediction of the threshold-channel theory. Unfortunately, our experimental setup does not allow us to measure [..143 ]the water discharge of individual channels. If the [..144 ]flow distributes itself evenly among the channels, though, we can approximate their individual discharges to a fraction of the total discharge.

To evaluate this approximation, we now analyze top-view pictures of our developing fans (about 15 pictures per run). We first divide the surface of each fan into five concentric [..145 ]bins, where we count the active channels and measure their widths (at least two cross sections per channel and per bin, Fig. 11). We then average the number of channels, and their width[..146 ], over experimental runs. The resulting quantities depend on the time of their measurement, and on the distance [..147 ]from the
* * *
[140]removed: The experiments of Stebbings (1963) suggest that sediment discharge causes a channel to widen, until it becomes unstable and breaks into a braid (Métivier et al., 2016b). The individual threads of a braid, in their turn, behave as threshold channels, both in laboratory flumes and in natural rivers (Reitz et al., 2014; Gaurav et al., 2015; Métivier et al., 2016a).

[141]removed: ramify

[142]removed: However, following the above contributions, we

[143]removed: their individual water discharges

[144]removed: water

[145]removed: strips

[146]removed: over

[147]removed: to

[revised manuscript text omitted]